# REWARD-AUGMENTED DATA ENHANCES DIRECT PREFERENCE ALIGNMENT OF LLMS

## ABSTRACT

Preference alignment in Large Language Models (LLMs) has significantly improved their ability to adhere to human instructions and intentions. However, existing direct alignment algorithms primarily focus on relative preferences and often overlook the qualitative aspects of responses, despite having access to preference data that includes reward scores from judge models during AI feedback. Striving to maximize the implicit reward gap between the chosen and the slightly inferior rejected responses can cause overfitting and unnecessary unlearning of the high-quality rejected responses. The unawareness of the reward scores also drives the LLM to indiscriminately favor the low-quality chosen responses and fail to generalize to responses with the highest rewards, which are sparse in data. To overcome these shortcomings, our study introduces reward-conditioned LLM policies that discern and learn from the entire spectrum of response quality within the dataset, helping extrapolate to more optimal regions. We propose an effective yet simple data relabeling method that conditions the preference pairs on quality scores to construct a reward-augmented dataset. This dataset is easily integrated with existing direct alignment algorithms and is applicable to any preference dataset. The experimental results across instruction-following benchmarks including AlpacaEval 2.0, MT-Bench, and Arena-Hard-Auto demonstrate that our approach consistently boosts the performance of DPO by a considerable margin across diverse models such as Zephyr, Mistral, Qwen2, Llama3.1, Gemma2, and SPPO. Additionally, on six academic benchmarks including GSM8K, GPQA, MUSR, TruthfulQA, BBH, and ARC, our method improves their average accuracy. When applying our method to on-policy data, the resulting DPO model outperforms various baselines and achieves state-of-the-art results on AlpacaEval 2.0. Through comprehensive ablation studies, we demonstrate that our method not only maximizes the utility of preference data but also mitigates the issue of unlearning, demonstrating its broad effectiveness beyond mere dataset expansion.

## 1 INTRODUCTION

Reinforcement Learning from Human Feedback (RLHF) has recently seen remarkable success in aligning Large Language Models (LLMs) to follow instructions with human intentions. In this approach, AI-generated feedback serves as a stand-in for human preferences, assessing and ranking responses to prompts to construct a preference dataset. This dataset is then utilized in preference optimization algorithms to fine-tune LLMs. Among them, direct preference alignment (Rafailov et al., 2024b; Azar et al., 2023; Zhao et al., 2023; Ethayarajh et al., 2024) that bypasses the need for an explicit reward model has garnered interest for their simplicity and cost efficiency. However, these algorithms mainly concern relative preferences and often overlook the quality of responses and their gaps, leading to limitations in their effectiveness.

Specifically, direct alignment algorithms such as DPO (Rafailov et al., 2024b) focus on maximizing the implicit reward difference between accepted and rejected responses. This approach can lead to overfitting, as high-quality but rejected responses are unnecessarily unlearned (Adler et al., 2024). Even worse, since the dataset provides only a sample estimate of true preferences, the rejected responses can actually be more aligned with human preferences than the accepted ones in expectation. Similarly, due to the unawareness of the responses' qualities, direct alignment will also result in the indiscriminate learning of the chosen responses, even when they are of low quality. As a result,

the directly aligned LLMs often struggle to differentiate between responses of varying quality and fail to generalize effectively to more optimal or the highest-reward responses that are sparse in the preference data, which is another limitation.

To address these issues, we propose learning reward-conditioned policies as a straightforward fix to the above issues. By optimizing the LLM to generate responses conditioning on their qualities, the model is allowed to discern and leverage patterns within responses of varied quality. As a result, learning from both chosen and rejected responses alleviates the issue of unnecessarily unlearning high-quality rejected responses; distinguishing between varying-quality chosen responses alleviates the issue of indiscriminately accepting low-quality ones. By identifying common patterns in responses of similar quality and distinguishing them from those of differing quality, the LLM becomes more adept at generalizing to more optimal responses that are sparse in data.

With this motivation, we introduce an effective yet simple data relabeling method to construct reward-augmented datasets. We define a goal-conditioned reward using an indicator function that compares the goal reward with the actual quality score, such as the reward value given by the judge model during AI feedback. This allows us to relabel each preference pair, generating two new pairs conditioned on the reward goals of both the chosen and rejected responses. The resulting augmented dataset, which contains these newly conditioned pairs, can enhance the performance of existing direct alignment algorithms. Our method can be applied to any preference dataset and followed by off-the-shelf direct alignment algorithms to boost their performance.

In experiments, we first apply our method on UltraFeedback (Cui et al., 2023) and perform DPO (Rafailov et al., 2024b) on this reward-augmented preference dataset by fine-tuning on various models, including Zephyr-7B-$\beta$ (Tunstall et al., 2023b), Mistral-7B-Instruct-v0.3 (Jiang et al., 2023a), Qwen2-7B-Instruct (Yang et al., 2024), Llama-3.1-8B-Instruct (Dubey et al., 2024), Gemma-2-9B-It (Team et al., 2024), and SPPO (Wu et al., 2024). The results show that our method consistently boosts the performance of these models as well as their DPO models by a large margin on instruction-following benchmarks such as AlpacaEval 2.0 (Dubois et al., 2024), MT-Bench (Zheng et al., 2024), and Arena-Hard-Auto (Li et al., 2024b). Our method also improves the average accuracy on a variety of academic benchmarks (GSM8K, GPQA, MUSR, TruthfulQA, BBH, and ARC). Moreover, our findings also demonstrate an improved utility of the preference data: a subsequent round of DPO using the reward-augmented data can still significantly enhance the model fine-tuned with DPO; relabeling the binarized preference dataset with the DPO implicit reward leads to further performance gains. Additional ablation studies also suggest that our method addresses the problem of unlearning and is superior not just due to the increased dataset size. When applied to on-policy data, our method enhances the DPO model, enabling it to surpass various baselines and achieve state-of-the-art performance on AlpacaEval 2.0.

## 2 BACKGROUND

Consider a language model $\pi \in \Delta_{\mathcal{Y}}^{\mathcal{X}}$ that takes the prompt $x \in \mathcal{X}$ as input and outputs the response $y \in \mathcal{Y}$, where $\mathcal{X}$ and $\mathcal{Y}$ are spaces of prompts and responses, respectively. Given the prompt $x \in \mathcal{X}$, a discrete probability distribution $\pi(\cdot \mid x) \in \Delta_{\mathcal{Y}}$ is generated, where $\Delta_{\mathcal{Y}}$ is the set of discrete distributions over $\mathcal{Y}$. We define the true human preference distribution as

$$p^*(y_1 \succ y_2 \mid x) := \mathbb{E}_h\big[\mathbb{1}(h \text{ prefers } y_1 \text{ over } y_2 \text{ given } x)\big],$$

where $h$ denotes the human rater and the expectation is over $h$ to account for the randomness of the human raters' choices. After pretraining and Supervised Fine-Tuning (SFT), Reinforcement Learning from Human or AI Feedback (Ouyang et al., 2022; Bai et al., 2022b) is typically employed to enhance the ability of the language model to follow instructions with human preferences.

**RL from AI Feedback (RLAIF).** The RLAIF framework involves two major steps: preference dataset construction with AI feedback and preference optimization. As a surrogate for human preference, AI feedback, including LLM-as-Judge (Zheng et al., 2024; Cui et al., 2023) and Reward-Model-as-Judge (Adler et al., 2024; Dong et al., 2024), can be used to rank responses and generate preference pairs. Specifically, consider the judge model $r(x, y) : \mathcal{X} \times \mathcal{Y} \to \mathbb{R}$ that outputs a scalar reward value representing the quality of $y$ under $x$. For each prompt $x \in \mathcal{X}$, two responses, $y_1$ and $y_2$, are independently sampled—either from the same reference model (Xiong et al., 2024; Wu

et al., 2024) or several different models (Zhu et al., 2023a; Zhang et al., 2024). Then $r(x, y_1)$ and $r(x, y_2)$ are evaluated to determine the preferred response $y_w = \text{argmax}_{y \in \{y_1, y_2\}} r(x, y)$ and dis-preferred response $y_l = \text{argmin}_{y \in \{y_1, y_2\}} r(x, y)$. By sampling responses and ranking them for a set of $N$ prompts, we get a preference dataset: $\mathcal{D}_N = \{(x^i, y_w^i, y_l^i)\}_{i=1}^N$. For the simplicity of our discussions, we assume that the reward function $r$ bounded in $[0, r_{\max}]$.

**Direct Alignment from Preference.** The objective for the LLM $\pi \in \Delta_{\mathcal{Y}}^{\mathcal{X}}$ is to maximize the KL-regularized expected reward. Recent works (Azar et al., 2023; Zhao et al., 2023; Tunstall et al., 2023b; Ethayarajh et al., 2024) proposed to align the LLM directly with the preference data by deriving the preference loss as a function of the LLM by the change of variables. Among them, the Direct Preference Optimization (DPO) (Rafailov et al., 2024b) loss has the following form:

$$\mathcal{L}_{\text{DPO}}(\pi; \mathcal{D}_N) = -\mathbb{E}_{(x, y_w, y_l) \sim \mathcal{D}_N} \left[ \log \sigma \left( \beta \log \frac{\pi(y_w \mid x)}{\pi_{\text{ref}}(y_w \mid x)} - \beta \log \frac{\pi(y_l \mid x)}{\pi_{\text{ref}}(y_l \mid x)} \right) \right],$$

where $\beta$ is a hyperparameter corresponding to the KL divergence regularization, $\sigma(\cdot)$ is the logistic function, and $\pi_{\text{ref}}$ is some reference LLM policy, such as the SFT model.

## 3 REWARD-CONDITIONING ADDRESSES LIMITATIONS OF DIRECT PREFERENCE ALIGNMENT

### 3.1 LIMITATIONS OF DIRECT ALIGNMENT FROM PREFERENCE

We will first demonstrate the limitations of vanilla direct alignment over the preference data.

**High-Quality Rejected Responses are Unnecessarily Suppressed.** The dataset $\mathcal{D}_N$ often contains preference pairs where the rejected response $y_l$ is only marginally worse than the chosen one $y_w$. Direct alignment algorithms, however, primarily focus on relative preferences and are unaware of the responses' quality values and gaps. Striving to maximize the reparameterized reward gap between the chosen and rejected responses will risk overfitting and unnecessary "unlearning", i.e., probability decrease, of high-quality responses, potentially diminishing the model's performance by discarding valuable alternatives. Furthermore, in such a finite data regime where only a sample estimate of the true preference is accessible, it can be very possible that $p^*(y_l \succ y_w \mid x) > 0.5$, i.e., $y_l$ is in fact more preferred than $y_w$ in expectation. This issue becomes even more pronounced when the preference data generated with the imperfect judge model is noisy.

We illustrate this limitation with the example in Table 1, where we define the maximum reward $r_{\max}$ as 10. For $\mathcal{D}_{N=1}$ that contains a single preference pair[1] with reward $r(x, y_1) = 9$ and $r(x, y_2) = 8$, the optimal policy learned from $\mathcal{D}_{N=1}$ is $\pi^*(y_1 \mid x) = 1$. This causes the model to avoid generating $y_2$, a response of nearly equivalent quality.

**Low-Quality Chosen Responses are Indiscriminately Learned.** For a similar reason, direct alignment algorithms also indiscriminately reinforce the chosen responses. As illustrated in Table 2, when $\mathcal{D}_{N=2}$ contains two preference pairs, where one of the chosen responses, $y_2$, is of low quality, $\pi^*$ still indiscriminately generates $y_2$ with an arbitrary probability $0 \leq a \leq 1$, i.e., $\pi^*(y_2 \mid x) = a$.

**Reward Sparsity.** Preference data often contains responses that, despite being preferred in pairwise comparisons, exhibit substantial variation in quality. As a result, the optimal responses—those associated with the highest reward value $r_{\max}$—are sparse in the dataset. Since direct alignment algorithms do not account for these reward values, the trained model struggles to differentiate between responses of varying quality and fails to generalize effectively to the sparse optimal responses.

### 3.2 REWARD-CONDITIONED POLICIES LEARN FROM THE FULL SPECTRUM OF RESPONSE QUALITY

A straightforward way to address the limitations of direct alignment algorithms—specifically, their inability to account for the quality of responses—is to optimize a reward-conditioned policy. In this

---

[1]For simplicity, we write $(x, y_w, y_l) \in \mathcal{D}_N$ as $y_w \succ y_l$.

| response | $y_1$ | $y_2$ |
|---|---|---|
| $r(x, y)$ | 9 | 8 |
| $\mathcal{D}_{N=1}$ | $\{y_1 > y_2\}$ | |
| $\pi^*(y \mid x)$ | 1 | 0 |
| $\pi^*(y \mid x, g=9)$ | 1 | 0 |
| $\pi^*(y \mid x, g=8)$ | 0 | 1 |

Table 1: High-quality rejected responses such as $y_2$ can be unnecessarily unlearned: $\pi^*(\cdot \mid x)$ deterministically generates $y_1$. Reward-conditioned policies learn both responses and are easier to generalize to $g = 10$ with the extracted features from $g = 8$ and $g = 9$.

| response | $y_1$ | $y_2$ | $y_3$ |
|---|---|---|---|
| $r(x, y)$ | 9 | 1 | 0 |
| $\mathcal{D}_{N=2}$ | $\{y_1 > y_3 , y_2 > y_3\}$ | | |
| $\pi^*(y \mid x)$ | $1-a$ | $a$ | 0 |
| $\pi^*(y \mid x, g=9)$ | 1 | 0 | 0 |
| $\pi^*(y \mid x, g=1)$ | 0 | 1 | 0 |
| $\pi^*(y \mid x, g=0)$ | 0 | 0 | 1 |

Table 2: Low-quality chosen responses such as $y_2$ can be learned: $\pi^*$ indiscriminately generates $y_1$ and $y_2$. Reward-conditioned policies distinguish the differences and learn the behaviors corresponding to different reward scores.

approach, the LLM policy is trained to generate responses corresponding to different reward values, enabling it to become aware of and adapt to these reward distinctions. By doing so, the LLM not only learns the patterns associated with the preferred responses but also retains the valuable information from the rejected ones, preventing the unlearning of high-quality rejected responses. For example, in Table 1, reward-conditioned policies learn to generate both $y_1$ and $y_2$, instead of unlearning $y_2$. This reward-based conditioning also enhances the model's ability to differentiate between responses of varying quality, even if both are preferred over a rejected alternative, as illustrated in Table 2. Besides, by extracting common patterns across responses with different quality levels, the LLM becomes more generalizable and is capable of generating the highest-quality responses with reward $r_{\max}$ (e.g., $r_{\max} = 10$), which are often sparse in the training preference data.

## 4 METHOD

With the above motivation, we propose a data relabeling method that constructs a reward-augmented dataset by conditioning the preference pairs on the reward values given by the judge model $r$. Specifically, we define the goal-conditioned reward function $R(x, y, g) = -(g - r(x, y))^2$ as a function of the reward function $r$. The objective of the reward-conditioned policy $\pi(y \mid x, g)$ is thus to minimize the square difference between the goal reward $g$ and the response reward $r(x, y)$, which is equivalent to maximizing the goal-conditioned reward $R(x, y, g)$, i.e.,

$$\min_\pi \mathbb{E}_{g, x \sim \mathcal{D}_N, y \sim \pi(\cdot|x,g)} \left[ (g - r(x, y))^2 \right] = \max_\pi \mathbb{E}_{g, x \sim \mathcal{D}, y \sim \pi(\cdot|x,g)} \left[ R(x, y, g) \right]. \quad (4.1)$$

To optimize the RHS of Equation (4.1), we first observe that under the new goal-conditioned reward metric $r$, for each preference pair $x^i, y_w^i, r_w^i, y_l^i, r_l^i$ in $\mathcal{D}_N$, we have

$$R(x, y_w^i, g = r_w^i) = 0 > R(x, y_l^i, g = r_w^i) = -(r_w^i - r_l^i)^2,$$
$$R(x, y_l^i, g = r_l^i) = 0 > R(x, y_w^i, g = r_l^i) = -(r_w^i - r_l^i)^2.$$

Thus, each pair can be relabeled to create two new preference pairs based on two distinct goals: when $g = r_w^i$, $y_w^i \succ y_l^i$; when $g = r_l^i$, $y_l^i \succ y_w^i$. Then any direct alignment algorithm can be applied to this new goal-conditioned preference dataset. Compared to fine-tuning on the original dataset $\mathcal{D}_N$, the model learns to capture not only desirable behaviors but also undesirable ones from the reward-augmented dataset. This approach helps identify patterns across high- and low-quality responses, enabling the LLMs to discern and learn from the entire spectrum of response quality and extrapolate to more optimal responses at inference time, by conditioning on higher reward goals.

We illustrate our method in Figure 1. For each preference pair with index $i$ in $\mathcal{D}_N$, two goals are defined, corresponding to the reward values of the chosen response $y_w^i$ and the rejected response $y_l^i$. Specifically, under the first goal $g = r_w^i$, the relabeled rewards are $R(x, y_w^i, g) = 0$ and $R(x, y_l^i, g) = -(r_w^i - r_l^i)^2$. The original ranking of responses remains the same, except that the LLM is preference optimized conditioned on $g = r_w^i$. Similarly, under the second goal $g = r_l^i$, the relabeled rewards are $R(x, y_l^i, g) = 0$ and $R(x, y_w^i, g) = -(r_w^i - r_l^i)^2$. Thus, the chosen and rejected

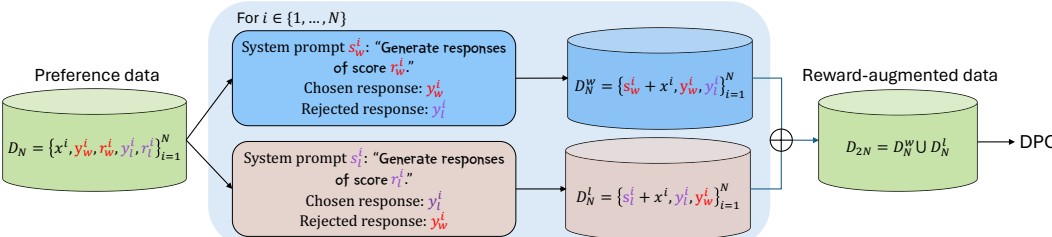

Figure 1: Construction of the reward-augmented preference dataset.

responses are reversed as $y_l^i$ and $y_w^i$, respectively. By generating preference pairs conditioned on the goal reward for both the chosen and rejected responses, we obtain a reward-augmented dataset of size $2N$. Finally, this new dataset can be used with any direct alignment algorithm, such as DPO.

In this work, we implement the reward-conditioned policy $\pi(y \mid x, g)$ as the LLM with a system prompt (or a prefix before the user prompt $x$ if system prompts are not supported by the LLM) such as "generate responses of score $g$". At inference time, the LLM is conditioned on the optimal goal $g^\star = r_{\max}$ that is the highest possible reward value, e.g., $g^\star = r_{\max} = 10$, to generate the responses.

We provide the following theoretical guarantees for our method (see A.4 for a formal description).

**Theorem 4.1** (Informal). Let $J(\pi) = \mathbb{E}_{x \sim d_0, y \sim \pi(\cdot|x, g^\star)}\big[R(x, y, g^\star)\big]$ be the performance measure, where $R$ denotes the ground-truth goal-conditioned reward function and $g^\star$ denotes the optimal goal. Under mild assumptions, the policy $\widehat{\pi}$ optimized from the reward-augmented DPO with a SFT regularizer satisfies that with probability at least $1 - \delta$,

$$J(\pi^*) - J(\widehat{\pi}) \le \sqrt{\frac{1}{N}} \cdot \Bigg\{ \frac{\sqrt{6}}{4}\big(1 + \exp(B)\big)^2 \big(\big(C_{\mu_{\bar{\mathcal{D}}}}(\mathcal{R}; \pi^\star, \pi_{\text{sft}})\big)^2 + 1\big)\iota$$

$$+ \mathbb{E}_{x \sim d_0}\Big[\text{KL}\big(\pi^\star(\cdot|x, g^\star)\|\pi_{\text{ref}}(\cdot|x, g^\star)\big)\Big] \Bigg\}, \tag{4.2}$$

where $\pi^* = \arg\max_\pi J(\pi)$ and $\iota = \sqrt{\log\big(\mathcal{N}_\varepsilon(\mathcal{R}, \|\cdot\|_\infty)/\delta\big)}$ with $\varepsilon = \big(6 \cdot (1 + e^B) \cdot N\big)^{-1}$. Here, $N$ denotes the number of preference pairs in $\mathcal{D}$, $B$ denotes the upper bound of the reward models, and the partial coverage coefficient $C_{\mu_{\bar{\mathcal{D}}}}(\mathcal{R}; \pi^\star, \pi_{\text{sft}})$ is defined in Assumption A.3.

The detailed proof is provided in A.5. The above theorem shows that our method attains global convergence to the optimal policy and the suboptimality decays at the order of $N^{-1/2}$ ($N$ denotes the size of the reward-augmented preference dataset), which provides a theoretical justification for the strong empirical performance of the introduced reward-augmented DPO. Unlike prior works on goal-conditioned RL with supervised learning (Yang et al., 2022; Ghosh et al., 2019), which typically establish weaker results such as local performance improvements or the optimization of a lower bound on $J(\pi)$, our analysis guarantees global convergence to the optimal policy. This distinction underscores the significance of integrating DPO-like methods with goal-conditioned approaches.

## 5 RELATED WORK

**Preference Dataset Construction.** In order for the LLMs to follow instructions and better align with human intents, it is common practice to build a preference dataset containing a set of prompts and a pair of responses for each prompt, whose qualities are ranked by humans (Ouyang et al., 2022) or judge models (Bai et al., 2022b). A popular pipeline (Cui et al., 2023; Tunstall et al., 2023b; Wang et al., 2024c; Ivison et al., 2023; Zhu et al., 2023a) for constructing offline (i.e., fixed) datasets involves sampling off-policy responses from *various* LLMs for each prompt in the hope to increase the response diversity. The preference data can also be generated online (Guo et al., 2024) or iteratively (Bai et al., 2022a; Xu et al., 2023; Gulcehre et al., 2023; Hoang Tran, 2024; Xiong et al., 2023; Dong et al., 2024; Calandriello et al., 2024; Rosset et al., 2024) by sampling and ranking on-policy responses from the training LLM. Recent works (Zhang et al., 2024; Cen et al., 2024; Xie et al., 2024) have also proposed systematically exploring the responses online and actively

eliciting the preference. The proposed method in this paper is orthogonal to the construction ways of the preference data and can be applied to any dataset created either off-policy or on-policy.

**Preference Optimization.** Preference optimization methods generally follow two approaches. The first involves learning a point-wise reward model, such as the Bradley-Terry model, and using RL algorithms like PPO (Schulman et al., 2017; Zheng et al., 2023; Xu et al., 2024b) or REIN-FORCE (Williams, 1992; Li et al., 2023; Ahmadian et al., 2024), to maximize the KL-regularized expected reward. The second approach is direct alignment (Rafailov et al., 2024b; Azar et al., 2023; Zhao et al., 2023; Ethayarajh et al., 2024; Liu et al., 2024), which gets rid of a separate reward model that is computationally costly to train. In this work, we mainly focus on the limitations of direct alignment algorithms, particularly their unawareness of the quality aspects of responses. For PPO-style alignment algorithms that fit and maximize an explicit reward, preference data is only used to learn the reward model, and policy training is performed in an online manner, where responses are sampled from the LLM and their reward values directly play a role during the RL optimization. This avoids drawbacks inherent to direct alignment methods, as detailed in Section 3.1.

**Conditional LLM Fine-Tuning.** Conditioning LLMs during training has proven effective for aligning responses with specific human objectives. SteerLM (Dong et al., 2023b; Wang et al., 2023b) extends SFT by conditioning the LLM on the multi-dimensional annotated attributes in data, such as humor and toxicity, in order to steer model responses with user customizability. Directional Preference Alignment (DPA) (Wang et al., 2024a) proposed a variant of rejection sampling fine-tuning (Yuan et al., 2023; Dong et al., 2023a) that conditions on the direction of the multi-objective reward, i.e., a user-dependent linear combination of the reward attributes (helpfulness and verbosity in their experiments), that represents diverse preference objectives. These methods aim to train a single LLM that can flexibly adjust to various user preference profiles. On the contrary, our method targets the limitations of direct alignment algorithms by introducing reward-augmented relabeling. This also differs from Conditioned-RLFT (Wang et al., 2023a), which leverages the data source information by learning a class-conditioned policy with RL-free supervised learning. Reward-aware Preference Optimization (RPO), introduced in Nemotron-4 (Adler et al., 2024), attempts to approximate the reward gap using the implicit reward and is motivated to resolve the unlearning issues of DPO, which our work also addresses. However, we show that more limitations beyond unlearning can be simply fixed with reward-conditioned LLMs and propose an easy-to-implement data relabeling method that integrates seamlessly with any direct alignment algorithm. Notably, Noise Contrastive Alignment (Chen et al., 2024) and Unified Language Model Alignment (Cai et al., 2023) introduce unified frameworks for alignment with binarized or reward datasets by leveraging (information) noise contrastive estimation and a hybrid of SFT with point-wise DPO, respectively. In contrast, our work focuses on addressing the limitations of direct alignment algorithms with data relabeling (on implicit-reward augmented binarized or reward datasets), and do not make algorithm changes. We compare with all the aforementioned methods in our experiments.

# 6 EXPERIMENTS

## 6.1 REWARD-AUGMENTED DATA BOOSTS DPO PERFORMANCE

We begin by conducting experiments to demonstrate that applying the proposed method to fixed offline preference datasets leads to consistent performance improvements in DPO.

**Setup.** We adopt the UltraFeedback (Cui et al., 2023) preference dataset containing reward values scored by GPT-4 (LLM-as-Judge) that is ranged between 1 and 10 for each of the preference pairs. Our method constructs reward-augmented data by conditioning on these judge values. We fine-tune on various open-weight LLMs, including Mistral-7B-Instruct-v0.3, Qwen2-7B-Instruct, Llama-3.1-8B-Instruct, Gemma-2-9B-It, and SPPO (fine-tuned from Gemma2-9B-It). We use the DPO implementation in the Huggingface Alignment Handbook (Tunstall et al.). The hyperparameters and prompts that we use are listed in Appendix B.1.

**Results.** We first report the performance of the trained models on instruction-following benchmarks that use LLM as a judge, including AlpacaEval 2.0 (Dubois et al., 2024), MT-Bench (Zheng et al., 2024), and Arena-Hard-Auto (Li et al., 2024b). The results are shown in Figure 2.

Across all instruction-following benchmarks, we observe that LLMs fine-tuned with DPO on the proposed reward-augmented data consistently outperform both their base models and those fine-tuned using DPO on the original UltraFeedback dataset by a considerable margin. Notably, direct alignment with the original preference data can sometimes degrade the performance of base models on specific benchmarks, such as Arena-Hard-Auto, which involves more complex reasoning tasks. In contrast, alignment using the reward-augmented data consistently yields superior results not only due to the improved style format gained from performing DPO on UltraFeedback.

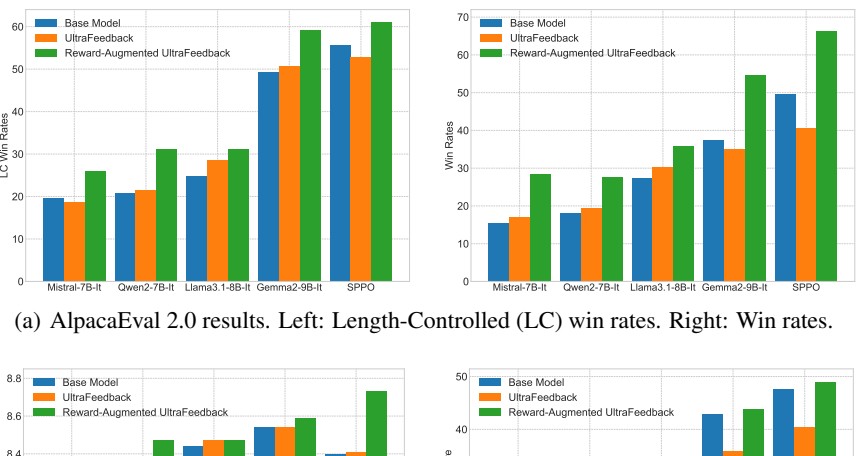

(a) AlpacaEval 2.0 results. Left: Length-Controlled (LC) win rates. Right: Win rates.

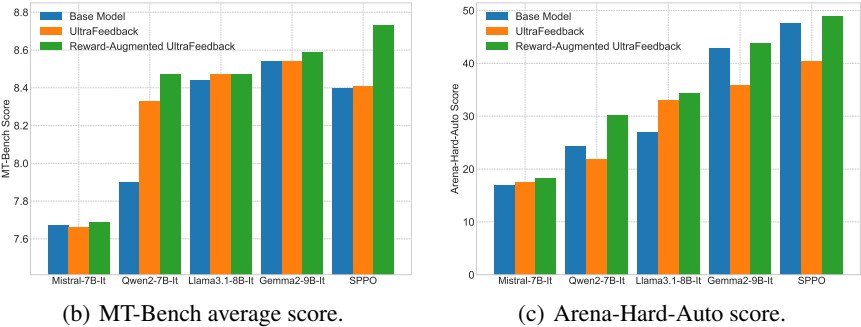

(b) MT-Bench average score.    (c) Arena-Hard-Auto score.

Figure 2: Performance of the base models, the models trained with DPO on UltraFeedback, and the models trained with DPO on reward-augmented ultrafeedback on AlpacaEval 2.0, MT-Bench, and Arena-Hard-Auto benchmarks. The complete table is deferred to Appendix B.2.

Besides, we also evaluate the models on academic multi-choice QA benchmarks, including GSM8K (Cobbe et al., 2021), GPQA (Rein et al., 2023), MUSR (Sprague et al., 2023), TruthfulQA (Lin et al., 2021), BBH (Suzgun et al., 2022), and ARC Challenge (Clark et al., 2018). To better reflect the capabilities of LLMs, we adopt various settings for these benchmarks, including zero-shot, few-shot, and few-shot Chain-of-Thought (CoT). The results are shown in Table 3.

It can be observed that performing DPO on the reward-augmented preference data leads to better average academic scores for most families of models compared to models fine-tuned on the original UltraFeedback dataset and the base models. Besides, we didn't observe severe alignment tax phenomenons (Askell et al., 2021; Noukhovitch et al., 2024; Li et al., 2024a) after DPO, and our method is able to improve the base models on most of the benchmarks.

## 6.2 ABLATION STUDIES

**Our Method Improves the Utility of Preference Data.** We provide two pieces of evidence that our method can get more juice out of the preference data compared to directly applying DPO. Firstly, we evaluate SPPO (Wu et al., 2024) fine-tuned with DPO on UltraFeedback (UF). The results are shown in Table 4. Since the SPPO model is already trained on UltraFeedback from Gemma-2-9B-It, an additional round of DPO training

|  | LC WR | WR | MT | Arena |
|---|---|---|---|---|
| SPPO | 55.60 | 49.61 | 8.40 | 47.6 |
| +DPO (UF) | 52.75 | 40.58 | 8.41 | 40.4 |
| +DPO (RA) | **60.97** | **66.41** | **8.73** | **49.0** |

Table 4: SPPO can be improved with DPO by performing reward augmentation on the same data.

| Model | GSM8K (8-s CoT) | GPQA (0-s) | MUSR (0-s) | TruthfulQA (0-s) | BBH (3-s) | ARC (25-s) | Average |
|---|---|---|---|---|---|---|---|
| Mistral-7B-Instruct-v0.3 | 52.39 | **30.62** | **47.35** | 59.71 | 46.64 | 58.53 | 49.21 |
| +DPO (UltraFeedback) | **53.22** | 28.94 | 47.35 | 64.74 | **47.46** | 60.32 | **50.34** |
| +DPO (Reward-Augmented) | 51.86 | 28.02 | 46.56 | **65.90** | 46.36 | **61.60** | 50.05 |
| Qwen2-7B-Instruct | 78.24 | 32.80 | 44.58 | 57.31 | **55.20** | 53.75 | 53.65 |
| +DPO (UltraFeedback) | 78.17 | 32.80 | 44.31 | **58.91** | 54.49 | 53.75 | 53.74 |
| +DPO (Reward-Augmented) | **81.05** | **32.97** | **45.77** | 57.99 | 54.94 | **54.52** | **54.54** |
| Llama-3.1-8B-Instruct | 76.72 | **33.89** | 39.95 | 54.00 | 50.74 | 55.38 | 51.78 |
| +DPO (UltraFeedback) | 78.47 | 33.72 | 43.39 | 56.61 | 51.31 | **57.51** | 53.50 |
| +DPO (Reward-Augmented) | **78.77** | 32.55 | **43.52** | **63.32** | **51.57** | 56.48 | **54.37** |
| Gemma-2-9B-It | 81.35 | **36.33** | 46.03 | 60.15 | 59.42 | 64.85 | 58.02 |
| +DPO (UltraFeedback) | 83.32 | 34.14 | 46.56 | 65.12 | 59.78 | **66.41** | 59.22 |
| +DPO (Reward-Augmented) | **83.62** | 35.74 | **48.15** | **65.27** | **59.82** | 65.87 | **59.75** |
| SPPO | 79.83 | 35.91 | 44.97 | 62.56 | **59.61** | 63.74 | 57.77 |
| +DPO (UltraFeedback) | **81.73** | 33.64 | 45.50 | 65.72 | 59.16 | **66.89** | 58.77 |
| +DPO (Reward-Augmented) | 80.67 | **36.16** | **48.68** | **67.39** | 58.88 | 65.53 | **59.55** |

Table 3: Performance comparison between the LLMs after DPO on UltraFeedback, on reward-augmented UltraFeedback, and their base models on academic multi-choice QA benchmarks in standard zero-shot, few-shot, and CoT settings. Here, n-s refers to n-shot, the **bold** texts represent the best results in each family of models.

with the same data significantly degrades its performance. In contrast, performing DPO on Reward-Augmented (RA) UltraFeedback results in substantial performance gains for SPPO, indicating that our method enhances the utility of the preference data.

The second evidence is that after DPO, the implicit reward can be used to relabel and augment the same preference data. Specifically, after training Qwen2-7B-Instruct with DPO on UltraFeedback, we leverage the resulting model $\pi_{\text{DPO}}$ to calculate the implicit reward for each prompt $x$ and response $y$, i.e., $\widehat{r} = \beta(\log \pi_{\text{DPO}}(y \mid x) - \log \pi_{\text{Qwen}}(y \mid x))$. Then we perform DPO on Qwen2-7B-Instruct using the Implicit-Reward-Augmented (IRA) UltraFeedback. The results are shown in Table 5. We observe that augmenting the data with the implicit reward from the DPO (UF) model leads to superior performance even compared to augmenting the data with reward scores from the LLM judge, i.e., DPO (RA). This result highlights that DPO does not fully exploit the potential of the data. Moreover, this ablation demonstrates that our method is compatible with binarized preference datasets that only contain chosen and rejected response pairs, bypassing the need for reward scores from judge models.

|  | LC WR | WR | MT | Arena |
|---|---|---|---|---|
| Qwen2-7B-It | 20.93 | 18.22 | 7.90 | 24.3 |
| +DPO (UF) | 21.46 | 19.35 | 8.33 | 21.9 |
| +DPO (RA) | 31.17 | 27.58 | 8.47 | **30.1** |
| +DPO (IRA) | **32.61** | **29.15** | **8.49** | 28.3 |

Table 5: A second round of DPO on the reward-augmented data, i.e., DPO (IRA), relabeled with the implicit reward from the DPO model at the first round, i.e., DPO (UF), significantly improves it. Our method helps get more juice out of the *binarized* (i.e., without judge model rewards) preference data.

**Reward-Augmented Data is Superior Not Just Due to Its Increased Size.** In this part, we show that the success of our method is not merely due to the increased size of the training dataset. To illustrate this, we perform DPO on the dataset where reward augmentation is applied to the first half of the UltraFeedback data, which we denote as DPO (Half RA). By doing so, the reward-augmented data is of the same size as the original dataset, but with only half of the prompts and the corresponding responses being utilized. It can be observed from Table 6 that DPO (Half RA) outperforms fine-tuning on the whole Ul-

|  | LC WR | WR | MT | Arena |
|---|---|---|---|---|
| Qwen2-7B-It | 20.93 | 18.22 | 7.90 | 24.3 |
| +DPO (UF) | 21.46 | 19.35 | 8.33 | 21.9 |
| +DPO (RA) | **31.17** | 27.58 | **8.47** | **30.1** |
| +DPO (Half RA) | 29.56 | **28.30** | 8.33 | 26.9 |
| Gemma-2-9B-It | 49.20 | 37.58 | 8.54 | 42.8 |
| +DPO (UF) | 50.70 | 35.02 | 8.54 | 35.8 |
| +DPO (RA) | **59.27** | **54.56** | 8.59 | **43.9** |
| +DPO (Half RA) | 53.12 | 43.74 | **8.66** | 41.3 |

Table 6: DPO trained on only half of the data with reward augmentation outperforms the baseline.

traFeedback (UF) by a large margin and achieves comparable performance to applying reward augmentation across the entire UF dataset, which is denoted as DPO (RA).

**Reward-Augmented Data Mitigates the Unlearning Issue.**
We first demonstrate that DPO suffers from the limitation of unnecessarily unlearning high-quality rejected responses, as discussed in Section 3.1. Specifically, on the test set of Ultra-Feedback, we calculate the log probability of each rejected response for the Qwen2-7B-Instruct model, its DPO (UF) model, and our method DPO (RA). In Figure 3, we plot the expected log probability for rejected responses with reward scores $\geq 5$. We find that DPO substantially decreases the probability of these high-quality rejected responses, confirming that the unlearning issue arises in practice. In contrast, our method alleviates this issue, although the probability is still slightly lower than the base model, which is proven to be the feature of DPO (Rafailov et al., 2024a; Zhang et al., 2024; Xu et al., 2024b).

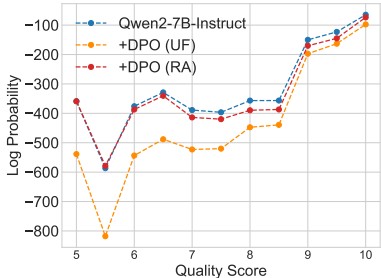

Figure 3: Our method helps mitigate the unlearning issue of DPO.

**Impact of the Accuracy of AI Feedback.** We consider the 19.8k prompts from a 1/3 subset of UltraFeedback following the setup from Snorkel (Hoang Tran, 2024). Five on-policy responses are first generated from Llama-3-8B-Instruct. An external reward model is followed to rank these responses. We choose the best and worst responses as the chosen and rejected ones. DPO is then performed on the resulting preference pairs and the reward-augmented pairs. To ablate how our method will be impacted by the accuracy of AI feedback, we experiment with two reward models as the ranker: PairRM (Jiang et al., 2023b) and ArmoRM (Wang et al., 2024b). PairRM is a small-sized (0.4B) pairwise reward model, while ArmoRM is a 8B model that is state-of-the-art on RewardBench (Lambert et al., 2024) and much stronger than PairRM. We implement a variant (denoted as RA+) of the proposed reward augmentation method that only conditions on the goal rewards of the chosen responses, not those of the rejected ones, leading to same-sized datasets.

| | Llama-3-8B-Instruct | PairRM (0.4B) | | | ArmoRM (8B) | |
| --- | --- | --- | --- | --- | --- | --- |
| | | DPO (UF) | DPO (RA+) | DPO (RA) | DPO (UF) | DPO (RA+) |
| LC WR | 22.92 | 41.76 | 44.72 | 48.20 | 42.32 | **48.73** |
| WR | 23.15 | 45.79 | 44.70 | **53.17** | 42.79 | 45.36 |

Table 7: Ablation on the impact of AI feedback quality on the AlpacaEval 2.0 benchmark.

The results in Table 7 demonstrate that training on augmented data conditioned on both chosen and rejected rewards is necessary for PairRM feedback, while relabeling with only the chosen rewards is sufficient to achieve strong performance for ArmoRM feedback. This aligns with our motivation outlined in Section 3.1: in noisy preference data, rejected responses may actually be of high quality, unlearning which can degrade performance. Similarly, low-quality chosen responses may also be reinforced. This issue does not arise with strong reward models that provide accurate preferences.

| | SLiC-HF | ORPO | CPO | RRHF | KTO | IPO | RPO | R-DPO | SimPO | Ours |
| --- | --- | --- | --- | --- | --- | --- | --- | --- | --- | --- |
| LC WR | 26.9 | 28.5 | 28.9 | 31.3 | 33.1 | 35.6 | 40.8 | 41.1 | 44.7 | **48.2** |
| WR | 27.5 | 27.4 | 32.2 | 28.4 | 31.8 | 35.6 | 41.7 | 37.8 | 40.5 | **53.2** |

Table 8: Comparison between our method, i.e., Llama-3-8B-Instruct+DPO (RA) and baselines fine-tuned on the same model and on-policy data ranked by PairRM.

Moreover, in Table 8, we compare our method and various baselines under the same setting on the AlpacaEval 2.0 benchmark, including SLiC-HF (Zhao et al., 2023), ORPO (Hong et al., 2024), CPO (Xu et al., 2024a), RRHF (Yuan et al., 2024), KTO (Ethayarajh et al., 2024), IPO (Azar et al., 2023), R-DPO (Park et al., 2024), and SimPO (Meng et al., 2024), where the results are from Meng et al. (2024), as well as the RPO (Adler et al., 2024) baseline that we implement. Our method outperforms the above algorithms by a considerable margin.

**Conditioning on Multi-Attribute Rewards Enables SOTA Models.** In previous parts, our

method is implemented by conditioning on the scalar reward values given by the judge models, either LLMs or reward models. We find that our approach is generalizable to settings of multi-dimensional rewards that correspond to different attributes, such as helpfulness and truthfulness. Specifically, we follow

|  | LC Win Rate | Win Rate | Avg. Len. |
|---|---|---|---|
| Ours | **56.57** | **52.19** | 1840 |
| SimPO | 53.70 | 47.50 | 1777 |
| OpenChat | 17.48 | 11.36 | 1362 |

Table 9: Our method trained with DPO achieves SOTA when conditioning on 5-dim rewards.

the setting from last part to construct the preference dataset by applying the ArmoRM reward model on the on-policy responses generated by Llama-3-8B-Instruct. Since ArmoRM is a multi-objective model that not only gives a scalar reward value but also predicts human-interpretable fine-grained attributes, we first select 5 attributes (namely complexity, instruction following, honesty, helpfulness, and intelligence depth) that have the highest average coefficients on the UltraFeedback data. Then we relabel the data by conditioning on the 5-dim reward and follow the implementation of using ArmoRM described in the last part. The resulting model achieves state-of-the-art within the Llama-3-8B-Instruct model family, surpassing the strong baselines including SimPO (Meng et al., 2024) that is trained also on on-policy data ranked by ArmoRM, and OpenChat (Wang et al., 2023a) fine-tuned with Conditioned-RLFT from the same Llama-3-8B-Instruct model.

**Comparison with Conditional Fine-Tuning Baselines.** We further compared with additional conditional post-training baselines on the offline UltraFeedback dataset (i.e., without on-policy data), including DPA (Wang et al., 2024a), SteerLM (Dong et al., 2023b), and (Info)NCA (Chen et al., 2024). Since both baselines aim to optimize a user-controllable attribute-conditioned LLM that is optimal under diverse preference profiles with different coefficients of the reward's attributes, in Figure 4, we plot the win rate curves of these methods under varying preference profiles, such as adjusting verbosity preferences as considered in Wang et al. (2024a). It can be observed that fine-tuned from Zephyr-SFT, our method achieves the best AlpacaEval 2.0 win rate. In addition to the comparison with the implemented RPO (Adler et al., 2024) in Table 8, we also report the performance of RPO fine-tuned on additional models including Qwen2-7B-Instruct and Gemma2-9B-It. As shown in Table 10, the implemented RPO is outperformed by our method across these models.

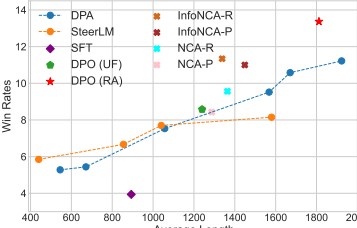

Figure 4: Comparison with DPA, SteerLM, and (Info)NCA.

|  | LC Win Rate | Win Rate | Avg. Len. |
|---|---|---|---|
| Qwen+RPO | 20.29 | 17.34 | 1704 |
| Qwen+DPO (RA) | **31.17** | **27.58** | 1789 |
| Gemma+RPO | 43.14 | 30.93 | 1413 |
| Gemma+DPO (RA) | **59.27** | **54.56** | 1872 |

Table 10: Comparison on AlpacaEval 2.0 between our method and RPO fine-tuned from the Qwen2-7B-Instruct and Gemma2-9B-It models. Our method consistently outperforms RPO across these fine-tuned models.

## 7 CONCLUSION

In this paper, we first investigate the limitations of direct alignment algorithms, which arise from focusing solely on relative preferences while neglecting the qualities of the responses and their gaps. Specifically, since many rejected responses are only slightly worse than the chosen ones, striving to maximize the reparameterized reward gap will cause overfitting and unnecessarily suppressing the high-quality rejected response. Moreover, the directly aligned LLMs often struggle to differentiate between responses of varying quality, indiscriminately learning the low-quality chosen responses and failing to generalize effectively to more optimal responses that are sparse in the preference data. To resolve the above limitations, we introduce a straightforward solution—learning reward-conditioned policies. By optimizing the LLM to generate responses conditioned on their qualities, it can better differentiate between quality levels and learn from the entire spectrum. Motivated by this, we propose a data relabeling method that constructs reward-augmented datasets by conditioning on the quality of responses as the goal quality. In experiments, we fine-tune various LLMs by applying DPO on our reward-augmented data. The results demonstrate that our approach consistently delivers significant performance improvements across various instruction-following benchmarks and increases the average accuracy on academic benchmarks.

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

# A  THEORY

In this section, we present the theoretical analysis for our proposed method.

## A.1  CONCEPTS

We provide some useful concepts for the simplicity of later discussions.

- Hellinger distance $D_{\text{Hellinger}}(p\|q)$ between two probability density functions $p$ and $q$ defined on $\mathcal{X}$ is defined as

$$D_{\text{Hellinger}}(p\|q) = \frac{1}{2}\int_{x\in\mathcal{X}}\left(\sqrt{p(x)} - \sqrt{q(x)}\right)^2 \mathrm{d}x.$$

- Total variation (TV) distance $D_{\text{TV}}(p\|q)$ between two probability density functions $p$ and $q$ defined on $\mathcal{X}$ is defined as

$$D_{\text{TV}}(p\|q) = \frac{1}{2}\int_{x\in\mathcal{X}}|p(x) - q(x)|\mathrm{d}x.$$

- Kullback–Leibler (KL) divergence $\text{KL}(p\|q)$ between two probability density functions $p$ and $q$ defined on $\mathcal{X}$ is defined as

$$\text{KL}(p\|q) = \int_{x\in\mathcal{X}}\log\left(\frac{p(x)}{q(x)}\right)p(x)\mathrm{d}x.$$

- We denote $\mathcal{N}_\epsilon(\mathcal{F}, \|\cdot\|_\infty)$ as the $\epsilon$-covering number (Zhou, 2002) for function class $\mathcal{F}$ under the infinity norm $\|\cdot\|_\infty$. Widely used in the theoretical analysis (Liu et al., 2024; Zhan et al., 2023), the $\epsilon$-covering number characterizes the complexity of the function class $\mathcal{F}$.

## A.2 THEORETICAL FORMULATION

**Goal-conditioned preference model.** Consider a language model $\pi \in \Delta_{\mathcal{Y}}^{\mathcal{X}}$ that takes the prompt $x \in \mathcal{X}$ as input and outputs the response $y \in \mathcal{Y}$, where $\mathcal{X}$ and $\mathcal{Y}$ are spaces of prompts and responses, respectively.Given the prompt $x \in \mathcal{X}$, a discrete probability distribution $\pi(\cdot \mid x) \in \Delta_{\mathcal{Y}}$ is generated, where $\Delta_{\mathcal{Y}}$ is the set of discrete distributions over $\mathcal{Y}$. We define the goal-conditioned reward function class as $\mathcal{R} \subset \{R(x, y, g) : \mathcal{X} \times \mathcal{Y} \times \mathcal{G} \mapsto \mathbb{R}\}$, where $\mathcal{G}$ is the goal space. The goal-conditioned Bradley-Terry model (Bradley & Terry, 1952) for annotations is described as

$$\mathbb{P}_R(y_1 \succ y_0 | x, y_1, y_0, g) = \frac{\exp(R(x, y_1, g))}{\exp(R(x, y_1, g)) + \exp(R(x, y_0, g))} = \sigma\big(R(x, y_1, g) - R(x, y_0, g)\big), \tag{A.1}$$

where $\sigma(z) = 1/(1 + \exp(-z))$ is the sigmoid function. For notational simplicity, we also denote that the reward is parameterized by $\theta \in \Theta$. We denote the corresponding negative log-likelihood function for $r$ for a reward-augmented preference dataset $\bar{\mathcal{D}} = \{(x^i, y_w^i, y_l^i, g^i)\}_{i=1}^N$ as

$$\mathcal{L}(R, \bar{\mathcal{D}}) = -\mathbb{E}_{(x, y_w, y_l, g) \sim \bar{\mathcal{D}}}\big[\log \sigma\big(R(x, y_w, g) - R(x, y_l, g)\big)\big], \tag{A.2}$$

where $y_w^i$ is preferred to $y_l^i$ by the annotation given the prompt $x^i$ and goal $g^i$ for any $i \in [N]$. For notational simplicity, we denote the DPO loss by

$$\mathcal{L}_{\text{DPO}}(\pi, \bar{\mathcal{D}}) = -\mathbb{E}_{(x, y_w, y_l, g) \sim \bar{\mathcal{D}}}\left[\log \sigma\left(\beta \log \frac{\pi(y_w \mid x, g)}{\pi_{\text{ref}}(y_w \mid x, g)} - \beta \log \frac{\pi(y_l \mid x, g)}{\pi_{\text{ref}}(y_l \mid x, g)}\right)\right]. \tag{A.3}$$

**Performance metric.** For the notational simplicity in the theoretical analysis, we denote by $R^\star$ the ground-truth goal-conditioned reward function. The alignment target is to maximize the expected true reward $R^\star$ conditioned on the optimal goal $g^\star \in \mathcal{G}$. Thus, we define the value function of any policy $\pi$ as

$$J(\pi) = \mathbb{E}_{x \sim d_0, y \sim \pi(\cdot | x, g^\star)}\big[R^\star(x, y, g^\star)\big]. \tag{A.4}$$

Here we allow the prompt distribution $d_0(\cdot)$ to be different from that of the offline dataset distribution $\mu_{\bar{\mathcal{D}}}(\cdot)$, but is assumed to be known. In the meanwhile, we consider the policies that share the same support as the reference policy $\pi_{\text{ref}}$ (Xiong et al., 2023), that is, we take a policy class $\Pi$ as

$$\Pi = \left\{\pi : \mathcal{X} \times \mathcal{G} \mapsto \Delta(\mathcal{A}) \,\middle|\, \text{Supp}(\pi(\cdot|x, g)) \subseteq \text{Supp}(\pi_{\text{ref}}(\cdot|x, g)), \ \forall(x, g) \in \mathcal{X} \times \mathcal{G}\right\}. \tag{A.5}$$

The performance gap of a learned policy $\widehat{\pi} \in \Pi$ w.r.t. any given optimal policy $\pi^\star$ is measured as

$$\text{Gap}^{\pi^\star}(\widehat{\pi}) = J(\pi^*) - J(\widehat{\pi}), \ \textit{given any optimal policy } \pi^\star \in \Pi, \tag{A.6}$$

One popular choice to define the optimal policy is to maximize the KL-regularized reward, i.e.,

$$\pi^\star = \underset{\pi \in \Pi}{\text{argmax}}\left[R^\star(x, y, g^\star) - \beta_0 \text{KL}\big(\pi(\cdot \mid x, g^\star) \| \pi_{\text{ref}}(\cdot \mid x, g^\star)\big)\right] \tag{A.7}$$

for a fixed $\beta_0 > 0$.

**Theoretical version of the reward-augmented DPO.** We formulate the theoretical version of the reward-augmented DPO in Algorithm 8, where we add a SFT regularizer on the empirical objective to handle the issue distribution shift and analyze the bound on the suboptimality (Liu et al., 2024; Cen et al., 2024). One simple choice to define SFT policy $\pi_{\text{SFT}}$ is to utilize the chosen labels in the original preference dataset $\mathcal{D}$, that is,

$$\pi_{\text{sft}} = \underset{\pi \in \Pi}{\text{argmax}} \, \mathbb{E}_{(x, y_w) \sim \mathcal{D}}[\log \pi(y_w \mid x, g^\star)]. \tag{A.8}$$

In practice, the goal relabeling distribution $g \sim p_{\mathcal{G}}(\cdot \mid x, y)$ is set to be a deterministic selection of the annotated reward of the chosen response, i.e., $g = r_{\text{RM}}(x, y)$ for any $i \in [N]$ and a given reward model $r_{\text{RM}}$. We also remark that the size of the reward-augmented preference dataset $\bar{\mathcal{D}}$ is $N = 2N_0$, where $N_0$ denotes the size of the original preference dataset $\mathcal{D}$.

---

**Algorithm 1** Theoretical Version of the Reward-Augmented DPO

---

1: **Input**: Preference dataset $\mathcal{D} = \{(x^i, y_w^i, y_l^i)\}_{i=1}^{N_0}$, parameters $\beta, \eta > 0$, reference policy $\pi_{\text{ref}}$, SFT policy $\pi_{\text{sft}}$ for the regularizer, and goal labeling distribution $p_{\mathcal{G}}$.
2: Initialize the reward-augmented preference dataset $\bar{\mathcal{D}} = \varnothing$.
3: **for** $i = 1, \ldots, N_0$ **do**
4:     Sample goal $g_w^i$ from $p_{\mathcal{G}}(\cdot \mid x^i, y_w^i)$ and update the reward-augmented preference dataset as $\bar{\mathcal{D}} \leftarrow \bar{\mathcal{D}} \cup \{(x^i, y_w^i, y_l^i, g_w^i)\}$.
5:     Sample goal $g_l^i$ from $p_{\mathcal{G}}(\cdot \mid x^i, y_l^i)$ and update the reward-augmented preference dataset as $\bar{\mathcal{D}} \leftarrow \bar{\mathcal{D}} \cup \{(x^i, y_l^i, y_w^i, g_l^i)\}$.
6: **end for**
7: Solve policy $\pi_{\widehat{\theta}}$ by optimizing the following objective

$$\min_{\theta \in \Theta} \left\{ \mathbb{E}_{x \sim d_0, y_0 \sim \pi_{\text{sft}}(\cdot | x, g^\star)} \left[ -\eta\beta \cdot \log(\pi_\theta(y_0 | x, g^\star)) \right] + \mathcal{L}_{\text{DPO}}(\pi_\theta, \bar{\mathcal{D}}) \right\} \qquad \text{(A.9)}$$

8: **Output**: Policy $\widehat{\pi} = \pi_{\widehat{\theta}}$.

---

### A.3    ASSUMPTIONS FOR THEORETICAL ANALYSIS

Similar to the theoretical analyses on offline RLHF (Liu et al., 2024; Cen et al., 2024), we provide the following assumptions.

**Assumption A.1** (True reward model). We assume that the true goal-conditioned reward model $R^\star \in \mathcal{R}$ for, and for any $R \in \mathcal{R}$ and $(x, y, g) \in \mathcal{X} \times \mathcal{A} \times \mathcal{G}$, it holds that $R(x, y, g) \in [-B/2, B/2]$ for a positive constant $B > 0$.

Assumption A.1 is standard in sample complexity analysis (Zhu et al., 2023b; Zhan et al., 2023; Ye et al., 2024) in RLHF.

**Assumption A.2** (Regularity). We assume that the reward model class $\mathcal{R}$, prompt space $\mathcal{X}$, and goal space $\mathcal{G}$ are convex and compact.

Assumption A.2 plays a role in establishing the equivalence between maximin and minimax optimizations. This assumption is naturally satisfied when considering a linear reward function (Zhu et al., 2023b; Xiong et al., 2023; Cen et al., 2024) of the form $R_\theta(x, y, g) = \varphi(x, y, g)^\top \theta$, where $\varphi$ represents a known feature map. More broadly, the assumption is also met by the class of Lipschitz continuous reward models.

**Assumption A.3** (Partial coverage coefficient). Given the optimal policy $\pi^\star \in \Pi$, the coverage coefficient of the population distribution $\mu_{\bar{\mathcal{D}}}$ of the reward-augmented preference dataset $\bar{\mathcal{D}}$ w.r.t. reward model class $\mathcal{R}$, optimal policy $\pi^\star$, and the SFT policy $\pi_{\text{sft}}$, denoted by $C_{\mu_{\bar{\mathcal{D}}}}(\mathcal{R}; \pi^\star, \pi_{\text{sft}})$, is defined as

$$\sup_{R \in \mathcal{R}} \frac{\mathbb{E}_{x \sim d_0, y_1 \sim \pi^\star(\cdot | x, g^\star), y_0 \sim \pi_{\text{sft}}(\cdot | x, g^\star)} \left[ (R^\star(x, y_1, g^\star) - R^\star(x, y_0, g^\star) - (R(x, y_1, g^\star) - R(x, y_0, g^\star))) \right]}{\sqrt{\mathbb{E}_{(x, y_w, y_l, g) \sim \mu_{\bar{\mathcal{D}}}} \left[ \left| (R(x, y_w, g) - R(x, y_l, g)) - (R(x, y_w, g) - R(x, y_l, g)) \right|^2 \right]}}. \qquad \text{(A.10)}$$

We assume that $C_{\mu_{\bar{\mathcal{D}}}}(\mathcal{R}; \pi^\star, \pi_{\text{sft}}) < +\infty$ for the given optimal policy $\pi^\star \in \Pi$.

Assumption A.3 characterizes how well the dataset $\bar{\mathcal{D}}$ covers the optimal policy $\pi^\star$ and the SFT policy $\pi_{\text{sft}}$ given the optimal goal $g^\star$, instead of covering all the policies in the policy class. That is the reason why we call this assumption "partial coverage". Different variants of partial coverage assumptions are posed in previous literature (Liu et al., 2024; Cen et al., 2024; Zhan et al., 2023; Xie et al., 2021) that study offline RLHF and RL to characterize the distribution shift between the optimal policy and the offline dataset distribution. We remark that the quantity $C_{\mu_{\bar{\mathcal{D}}}}(\mathcal{R}; \pi^\star, \pi_{\text{sft}})$ is upper bounded by the density ratio $\|d_0(\cdot) \otimes \pi^\star(\cdot | \cdot, g^\star) \otimes \pi_{\text{sft}}(\cdot | \cdot, g^\star) / \mu_{\bar{\mathcal{D}}}(\cdot, \cdot, \cdot, g^\star)\|_\infty$.

## A.4 THEORETICAL RESULTS

Under assumptions introduced before, we are ready to give the theoretical result for Algorithm 1 in the following theorem.

**Theorem A.4** (Suboptimality of Algorithm 8)**.** Taking the policy class $\Pi$ as (A.5), supposing that Assumptions A.1, A.2, and A.3 hold, and assuming that the reward model class $\mathcal{R}$ has a finite $\varepsilon$-covering number under $\|\cdot\|_\infty$-norm $\mathcal{N}_\varepsilon(\mathcal{R}, \|\cdot\|_\infty) < +\infty$ with $\varepsilon = (6 \cdot (1 + e^B) \cdot N)^{-1}$. Setting

$$\eta = (1 + \exp(B))^{-2} \cdot \sqrt{24 \log\left(\mathcal{N}_\varepsilon(\mathcal{R}, \|\cdot\|_\infty)/\delta\right)/N}, \quad \beta = 1/\sqrt{N}$$

in Algorithm 1. Then the output policy $\widehat{\pi}$ of Algorithm 1 satisfies that with probability at least $1 - \delta$,

$$\mathrm{Gap}^{\pi^\star}(\widehat{\pi}) \le \sqrt{\frac{1}{N}} \cdot \left\{ \frac{\sqrt{6}}{4} \left(1 + \exp(B)\right)^2 \left(\left(C_{\mu_{\bar{\mathcal{D}}}}(\mathcal{R}; \pi^\star, \pi_{\mathrm{sft}})\right)^2 + 1\right)\iota \right.$$
$$\left. + \mathbb{E}_{x \sim d_0}\left[\mathrm{KL}\left(\pi^\star(\cdot|x, g^\star) \| \pi_{\mathrm{ref}}(\cdot|x, g^\star)\right)\right] \right\}, \tag{A.11}$$

where $\iota = \sqrt{\log\left(\mathcal{N}_\varepsilon(\mathcal{R}, \|\cdot\|_\infty)/\delta\right)}$ with $\varepsilon = (6 \cdot (1 + e^B) \cdot N)^{-1}$. Here, $N$ denotes the number of preference pairs in $\mathcal{D}$, $R$ denotes the upper bound of the reward models, and the partial coverage coefficient $C_{\mu_{\bar{\mathcal{D}}}}(\mathcal{R}; \pi^\star, \pi_{\mathrm{sft}})$ is defined in Assumption A.3.

The detailed proof is provided in Appendix A.5. Theorem A.4 shows that our proposed reward-augmented DPO (Algorithm 1) can attain global convergence to the optimal policy and the sub-optimality decays at the order of $N^{-1/2}$ ($N$ denotes the size of the reward-augmented preference dataset). Theorem A.4 provides a theoretical justification for the strong empirical performance of the reward-augmented DPO introduced in this paper. Unlike prior works on goal-conditioned reinforcement learning with supervised learning (Yang et al., 2022; Ghosh et al., 2019), which typically establish weaker results such as local performance improvements or the optimization of a lower bound on $J(\pi)$, our analysis guarantees global convergence to the optimal policy. This distinction underscores the significance of integrating DPO-like methods with goal-conditioned approaches.

## A.5 PROOF OF THEOREM A.4

**Bridge Algorithm 1 to the maximin optimization.**    Motivated by Liu et al. (2024), we transform the optimization objective in Algorithm 1 to a minimax optimization objective, and then to a maximum optimization objective, where the maximum optimization objective can be analyzed with tools in RL analysis.

Define the objective function $\phi(\pi, R)$ as

$$\phi(\pi, R) = \eta \cdot \mathbb{E}_{\substack{x \sim d_0, y_1 \sim \pi(\cdot|x, g^\star) \\ y_0 \sim \pi_{\mathrm{sft}}(\cdot|x, g^\star)}}\left[R(x, y_1, g^\star) - R(x, y_0, g^\star)\right.$$
$$\left. - \beta \cdot D_{\mathrm{KL}}\left(\pi(\cdot|x, g^\star) \| \pi_{\mathrm{ref}}(\cdot|x, g^\star)\right)\right] + \mathcal{L}(R, \bar{\mathcal{D}}). \tag{A.12}$$

First, we prove that the derived policy $\widehat{\pi}$ from Algorithm 1 satisfies

$$\widehat{\pi} \in \operatorname*{argmax}_{\pi \in \Pi} \phi(\widehat{R}, \pi), \quad \text{where} \quad \widehat{R} \in \operatorname*{argmin}_{R \in \mathcal{R}} \max_{\pi \in \Pi} \phi(\pi, R). \tag{A.13}$$

By the definition of the optimization objective $\phi(\pi, R)$ in (A.12), we have

$$\min_{R \in \mathcal{R}} \max_{\pi \in \Pi} \phi(\pi, R) = \min_{R \in \mathcal{R}}\left\{\eta \cdot \max_{\pi \in \Pi}\left\{\mathbb{E}_{x \sim d_0, y_1 \sim \pi(\cdot|x, g^\star)}\left[R(x, y_1, g^\star) - \beta \cdot \mathrm{KL}\left(\pi(\cdot|x, g^\star) \| \pi_{\mathrm{ref}}(\cdot|x, g^\star)\right)\right]\right\}\right.$$
$$\left. - \eta \cdot \mathbb{E}_{x \sim d_0, y_0 \sim \pi_{\mathrm{sft}}(\cdot|x, g^\star)}\left[R(x, y_0, g^\star)\right] + \mathcal{L}(R, \bar{\mathcal{D}})\right\}. \tag{A.14}$$

Then, we apply the following lemma to solve the inner maximization problem in (A.14).

**Lemma A.5** (Oracle optimal KL-regularized policy). Given any reward model $R \in \mathcal{R}$, the optimal policy $\pi_R$ to the maximization problem

$$\max_{\pi \in \Pi} \left\{ \mathbb{E}_{x \sim d_0, y \sim \pi(\cdot | x, g^\star)} \left[ R(x, y, g^\star) - \beta \cdot \mathrm{KL}\big(\pi(\cdot | x, g^\star) \| \pi_{\mathrm{ref}}(\cdot | x, g^\star)\big) \right] \right\}. \tag{A.15}$$

is given by

$$\pi_R(\cdot | x, g) = \frac{1}{Z_R(x, g)} \cdot \pi^{\mathrm{ref}}(\cdot | x, g) \cdot \exp\big(\beta^{-1} R(x, \cdot, g)\big), \tag{A.16}$$

$$Z_R(x, g) = \int_{y \in \mathcal{Y}} \exp\big(\beta^{-1} R(x, y, g)\big) \, \mathrm{d}\pi_{\mathrm{ref}}(y | x, g),$$

and correspondingly the optimal value of (A.15) is given by $(A.15) = \mathbb{E}_{x \sim d_0}[\beta \cdot \log(Z_R(x, g^\star))]$.

*Proof of Lemma A.5.* See the proof in Lemma 4.2 of Liu et al. (2024). $\qquad\square$

By Lemma A.5 and (A.14), we have

$$\min_{R \in \mathcal{R}} \max_{\pi \in \Pi} \phi(\pi, R) = \min_{R \in \mathcal{R}} \left\{ \beta\eta \cdot \log(Z_R(x, g^\star)) - \eta \cdot \mathbb{E}_{x \sim d_0, y_0 \sim \pi_{\mathrm{sft}}(\cdot | x, g^\star)} \left[ R(x, y_0, g^\star) \right] + \mathcal{L}(R, \bar{\mathcal{D}}) \right\}. \tag{A.17}$$

From Lemma A.5, we know that given any reward model $R \in \mathcal{R}$, we can reparameterize it via its corresponding optimal goal-conditioned KL-regularized policy $\pi_R$ (Rafailov et al., 2024b), that is,

$$R(x, \cdot, g) = \beta \cdot \log\left(\frac{\pi_R(\cdot | x, g)}{\pi^{\mathrm{ref}}(\cdot | x, g)}\right) + \beta \cdot \log(Z_R(x, g)). \tag{A.18}$$

Plugging (A.18) into (A.19), we show that the optimization problem in Algorithm 1 relates to the minimax optimization problem on $\phi(\pi, R)$:

$$\min_{R \in \mathcal{R}} \max_{\pi \in \Pi} \phi(\pi, R) = \min_{R \in \mathcal{R}} \left\{ \eta\beta \cdot \mathbb{E}_{x \sim d_0, y_0 \sim \pi_{\mathrm{sft}}(\cdot | x, g^\star)} \left[ \log\left(\frac{\pi_R(y_0 | x, g^\star)}{\pi_{\mathrm{ref}}(y_0 | x, g^\star)}\right) \right] + \mathcal{L}_{\mathrm{DPO}}(\pi_R, \bar{\mathcal{D}}) \right\}$$

$$= \min_{R \in \mathcal{R}} \left\{ \eta\beta \cdot \mathbb{E}_{x \sim d_0, y_0 \sim \pi_{\mathrm{sft}}(\cdot | x, g^\star)} \left[ \log\left(\pi_R(y_0 | x, g^\star)\right) \right] + \mathcal{L}_{\mathrm{DPO}}(\pi_R, \bar{\mathcal{D}}) \right\}. \tag{A.19}$$

where the first equality uses the definition of DPO loss $\mathcal{L}_{\mathrm{DPO}}$ in (A.3). Since we know that $\widehat{\pi} \in \mathrm{argmax}_{\pi \in \Pi} \phi(\widehat{R}, \pi)$ and $\widehat{r}$ solves the minimization problem in (A.19), we know that $\widehat{\pi} = \pi_{\widehat{R}}$ by Lemma A.5.

Next, we show that the minimization problem $\phi(\pi, R)$ can be equivalently transformed into a maximization problem. Specifically, we will prove that the output policy $\widehat{\pi}$ for the Algorithm 1 satisfies

$$\widehat{\pi} \in \mathrm{argmax}_{\pi \in \Pi} \min_{R \in \mathcal{R}} \phi(\pi, R), \tag{A.20}$$

which is implied by the following theorem.

**Theorem A.6.** For the policy class $\Pi$ defined in (A.5) and the reward model class $\mathcal{R}$ satisfying Assumption A.2, consider the following policy defined as

$$\pi_{\widehat{R}} \in \mathrm{argmax}_{\pi \in \Pi} \phi(\widehat{R}, \pi), \quad \text{where} \quad \widehat{R} \in \mathrm{argmin}_{R \in \mathcal{R}} \max_{\pi \in \Pi} \phi(\pi, R). \tag{A.21}$$

Then the policy $\pi_{\widehat{R}}$ also solves the following maximin optimization:

$$\pi_{\widehat{R}} \in \mathrm{argmax}_{\pi \in \Pi} \min_{R \in \mathcal{R}} \phi(\pi, R). \tag{A.22}$$

*Proof.* Under Assumption A.1, we know that $\phi(\pi, R)$ is convex for $R \in \mathcal{R}$ and strongly concave for $\pi \in \Pi$. Applying Theorem 5.6 in Liu et al. (2024), we prove Theorem A.6. $\qquad\square$

**Suboptimality Decomposition.** By the definitions of the optimization objective $\phi(\pi, R)$ in (A.12) and the suboptimality gap of $\widehat{\pi}$ w.r.t. $\pi^\star$ in (A.6), we decompose the gap as

$$\text{Gap}^{\pi^\star}(\widehat{\pi})$$
$$= \mathbb{E}_{x \sim d_0, y \sim \pi^\star(\cdot|x,g^\star)}\big[R^\star(x,y,g^\star)\big] - \mathbb{E}_{x \sim d_0, a \sim \widehat{\pi}(\cdot|x,g^\star)}\big[R^\star(x,y,g^\star)\big]$$
$$= \mathbb{E}_{x \sim d_0, y_1 \sim \pi^\star(\cdot|x,g^\star), y_0 \sim \pi_{\text{sft}}(\cdot|x,g^\star)}\Big[R^\star(x,y_1,g^\star) - R^\star(x,y_0,g^\star) - \beta \cdot \text{KL}\big(\pi^\star(\cdot|x,g^\star)\|\pi_{\text{ref}}(\cdot|x,g^\star)\big)\Big]$$
$$\quad - \eta^{-1} \cdot \min_{R \in \mathcal{R}} \phi(\widehat{\pi}, R) + \eta^{-1} \cdot \min_{R \in \mathcal{R}} \phi(\widehat{\pi}, R)$$
$$\quad - \mathbb{E}_{x \sim d_0, y_1 \sim \widehat{\pi}(\cdot|x,g^\star), y_0 \sim \pi_{\text{sft}}(\cdot|x,g^\star)}\Big[R^\star(x,y_1,g^\star) - R^\star(x,y_0,g^\star) - \beta \cdot \text{KL}\big(\widehat{\pi}(\cdot|x,g^\star)\|\pi_{\text{ref}}(\cdot|x,g^\star)\big)\Big]$$
$$\quad + \beta \cdot \mathbb{E}_{x \sim d_0}\Big[\text{KL}\big(\pi^\star(\cdot|x,g^\star)\|\pi_{\text{ref}}(\cdot|x,g^\star)\big) - \text{KL}\big(\widehat{\pi}(\cdot|x,g^\star)\|\pi_{\text{ref}}(\cdot|x,g^\star)\big)\Big]$$
$$:= \text{Term (A)} + \text{Term (B)} + \text{Term (C)}, \tag{A.23}$$

where we abbreviate Term (A), Term (B), and Term (C) as follows

$$\text{Term (A)} = -\eta^{-1} \cdot \min_{R \in \mathcal{R}} \phi(\widehat{\pi}, R)$$
$$= \mathbb{E}_{x \sim d_0, y_1 \sim \pi^\star(\cdot|x,g^\star), y_0 \sim \pi_{\text{sft}}(\cdot|x,g^\star)}\Big[R^\star(x,y_1,g^\star) - R^\star(x,y_0,g^\star) - \beta \cdot \text{KL}\big(\pi^\star(\cdot|x,g^\star)\|\pi_{\text{ref}}(\cdot|x,g^\star)\big)\Big], \tag{A.24}$$

$$\text{Term (B)} = \eta^{-1} \cdot \min_{R \in \mathcal{R}} \phi(\widehat{\pi}, R)$$
$$\quad - \mathbb{E}_{x \sim d_0, y_1 \sim \widehat{\pi}(\cdot|x,g^\star), y_0 \sim \pi_{\text{sft}}(\cdot|x,g^\star)}\Big[R^\star(x,y_1,g^\star) - R^\star(x,y_0,g^\star) - \beta \cdot \text{KL}\big(\widehat{\pi}(\cdot|x,g^\star)\|\pi_{\text{ref}}(\cdot|x,g^\star)\big)\Big], \tag{A.25}$$

and

$$\text{Term (C)} = \beta \cdot \mathbb{E}_{x \sim d_0}\Big[\text{KL}\big(\pi^\star(\cdot|x,g^\star)\|\pi_{\text{ref}}(\cdot|x,g^\star)\big) - \text{KL}\big(\widehat{\pi}(\cdot|x,g^\star)\|\pi_{\text{ref}}(\cdot|x,g^\star)\big)\Big]. \tag{A.26}$$

In the following, we bound Term (A) and Term (B) respectively.

**Analysis of Term (A) in** (A.23). Note that

$$\text{Term (A)}$$
$$= \mathbb{E}_{x \sim d_0, y_1 \sim \pi^\star(\cdot|x,g^\star), y_0 \sim \pi_{\text{sft}}(\cdot|x,g^\star)}\Big[R^\star(x,y_1,g^\star) - R^\star(x,y_0,g^\star) - \beta \cdot \text{KL}\big(\pi^\star(\cdot|x,g^\star)\|\pi_{\text{ref}}(\cdot|x,g^\star)\big)\Big]$$
$$\quad - \eta^{-1} \cdot \min_{R \in \mathcal{R}} \phi(\widehat{\pi}, R)$$
$$\leq \mathbb{E}_{x \sim d_0, y_1 \sim \pi^\star(\cdot|x,g^\star), y_0 \sim \pi_{\text{sft}}(\cdot|x,g^\star)}\Big[R^\star(x,y_1,g^\star) - R^\star(x,y_0,g^\star) - \beta \cdot \text{KL}\big(\pi^\star(\cdot|x,g^\star)\|\pi_{\text{ref}}(\cdot|x,g^\star)\big)\Big]$$
$$\quad - \eta^{-1} \cdot \min_{R \in \mathcal{R}} \phi(\pi^\star, R)$$
$$= \max_{R \in \mathcal{R}} \Bigg\{ \mathbb{E}_{x \sim d_0, y_1 \sim \pi^\star(\cdot|x,g^\star), y_0 \sim \pi_{\text{sft}}(\cdot|x,g^\star)}\Big[\big(R^\star(x,y_1,g^\star) - R^\star(x,y_0,g^\star)\big) - \big(R(x,y_1,g^\star) - R(x,y_0,g^\star)\big)\Big]$$
$$\quad - \eta^{-1} \cdot \mathcal{L}(R, \bar{\mathcal{D}}) \Bigg\}, \tag{A.27}$$

where the inequality follows the fact that $\widehat{\pi}$ solves the maxmin optimization problem in (A.20).

**Analysis of Term (B) in** (A.23). Note that

Term (B)

$$= \eta^{-1} \cdot \min_{R \in \mathcal{R}} \phi(\widehat{\pi}, R)$$

$$- \mathbb{E}_{x \sim d_0, y_1 \sim \widehat{\pi}(\cdot|x,g^\star), y_0 \sim \pi_{\mathrm{sft}}(\cdot|x,g^\star)} \Big[ R^\star(x, y_1, g^\star) - R^\star(x, y_0, g^\star) - \beta \cdot \mathrm{KL}\big(\widehat{\pi}(\cdot|x,g^\star)\|\pi_{\mathrm{ref}}(\cdot|x,g^\star)\big) \Big]$$

$$\leq \mathbb{E}_{x \sim d_0, y_1 \sim \widehat{\pi}(\cdot|x,g^\star), y_0 \sim \pi_{\mathrm{sft}}(\cdot|x,g^\star)} \Big[ R^\star(x, y_1, g^\star) - R^\star(x, y_0, g^\star) - \beta \cdot \mathrm{KL}\big(\widehat{\pi}(\cdot|x,g^\star)\|\pi_{\mathrm{ref}}(\cdot|x,g^\star)\big) \Big]$$

$$+ \eta^{-1} \cdot \mathcal{L}(R^\star, \bar{\mathcal{D}})$$

$$- \mathbb{E}_{x \sim d_0, y_1 \sim \widehat{\pi}(\cdot|x,g^\star), y_0 \sim \pi_{\mathrm{sft}}(\cdot|x,g^\star)} \Big[ R^\star(x, y_1, g^\star) - R^\star(x, y_0, g^\star) - \beta \cdot \mathrm{KL}\big(\widehat{\pi}(\cdot|x,g^\star)\|\pi_{\mathrm{ref}}(\cdot|x,g^\star)\big) \Big]$$

$$= \eta^{-1} \cdot \mathcal{L}(R^\star, \bar{\mathcal{D}}), \tag{A.28}$$

where the inequality uses the fact that $R^\star \in \mathcal{R}$ by Assumption A.1 and the definition of the optimization objective in (A.12).

**Concluding the remaining proof.** Combining (A.23), (A.27), and (A.28), we have

$$\mathrm{Gap}_\beta^{\pi^\star}(\widehat{\pi}) = \text{Term (A)} + \text{Term (B)} + \text{Term (C)}$$

$$\leq \max_{R \in \mathcal{R}} \Bigg\{ \mathbb{E}_{\substack{x \sim d_0, y_1 \sim \pi^\star(\cdot|x,g^\star),\\ y_0 \sim \pi_{\mathrm{sft}}(\cdot|x,g^\star)}} \Big[ \big(R^\star(x, y_1, g^\star) - R^\star(x, y_0, g^\star)\big) - \big(R(x, y_1, g^\star) - R(x, y_0, g^\star)\big) \Big]$$

$$+ \eta^{-1} \cdot \Big( \mathcal{L}(R^\star, \bar{\mathcal{D}}) - \mathcal{L}(R, \bar{\mathcal{D}}) \Big) \Bigg\}$$

$$+ \beta \cdot \mathbb{E}_{x \sim d_0} \Big[ \mathrm{KL}\big(\pi^\star(\cdot|x,g^\star)\|\pi_{\mathrm{ref}}(\cdot|x,g^\star)\big) - \mathrm{KL}\big(\widehat{\pi}(\cdot|x,g^\star)\|\pi_{\mathrm{ref}}(\cdot|x,g^\star)\big) \Big]. \tag{A.29}$$

Next, we upper bound the right-hand side of (A.29) by relating the negative log-likelihood loss difference term to the reward difference term. Recall the definition of the goal-conditioned preference model $\mathbb{P}_R$ in (A.1). Applying Lemma A.7 to give an upper bound of the difference of the negative log-likelihood loss and setting $\varepsilon = (6 \cdot (1 + e^B) \cdot N)^{-1}$, it holds with probability at least $1 - \delta$ and for any reward model $R \in \mathcal{R}$ that

$$\mathcal{L}(R^\star, \bar{\mathcal{D}}) - \mathcal{L}(R, \bar{\mathcal{D}})$$

$$\leq -2 \cdot \mathbb{E}_{(x,y_1,y_0,g) \sim \mu_{\bar{\mathcal{D}}}} \Big[ D_{\mathrm{Hellinger}}^2 \big(\mathbb{P}_{R^\star}(\cdot|x, y_1, y_0, g)\|\mathbb{P}_R(\cdot|x, y_1, y_0, g)\big) \Big]$$

$$+ \frac{3}{N} \cdot \log\left( \frac{\mathcal{N}_\varepsilon(\mathcal{R}, \|\cdot\|_\infty)}{\delta} \right), \tag{A.30}$$

where $\mathcal{N}_\varepsilon(\mathcal{R}, \|\cdot\|_\infty)$ denotes the $\varepsilon$-covering number (Zhou, 2002) of the reward model class $\mathcal{R}$. By the relationship between the Hellinger distance and TV distance, we have

$$D_{\mathrm{Hellinger}}^2\big(\mathbb{P}_{R^\star}(\cdot|x, y_1, y_0, g)\|\mathbb{P}_R(\cdot|x, y_1, y_0, g)\big) \geq D_{\mathrm{TV}}^2\big(\mathbb{P}_{R^\star}(\cdot|x, y_1, y_0, g)\|\mathbb{P}_R(\cdot|x, y_1, y_0, g)\big),$$

By the definition of the goal-conditioned preference model $\mathbb{P}_R$ in (A.1), we have

$$D_{\mathrm{TV}}\big(\mathbb{P}_{R^\star}(\cdot|x, y_1, y_0, g)\|\mathbb{P}_R(\cdot|x, y_1, y_0, g)\big)$$

$$= \frac{1}{2} \cdot \Big| \sigma\big(R^\star(x, y_1, g^\star) - R^\star(x, y_0, g^\star)\big) - \sigma\big(R(x, y_1, g^\star) - R(x, y_0, g^\star)\big) \Big|$$

$$+ \frac{1}{2} \cdot \Big| \sigma\big(R^\star(x, y_0, g^\star) - R^\star(x, y_1, g^\star)\big) - \sigma\big(R(x, y_0, g^\star) - R(x, y_1, g^\star)\big) \Big|$$

$$= \Big| \sigma\big(R^\star(x, y_1, g^\star) - R^\star(x, y_0, g^\star)\big) - \sigma\big(R(x, y_1, g^\star) - R(x, y_0, g^\star)\big) \Big|, \tag{A.31}$$

where the second equality uses the fact that $\sigma(-z) = 1 - \sigma(z)$. Applying Lemma A.8 and the condition that $R(x, y, g) \in [B/2, B/2]$ for any $(x, y, R, g) \in \mathcal{X} \times \mathcal{A} \times \mathcal{R} \times \mathcal{G}$ in Assumption A.1, we have

$$\Big| \sigma\big(R^\star(x, y_1, g^\star) - R^\star(x, y_0, g^\star)\big) - \sigma\big(R(x, y_1, g^\star) - R(x, y_0, g^\star)\big) \Big|$$

$$\geq \kappa \cdot \Big| \big(R^\star(x, y_1, g^\star) - R^\star(x, y_0, g^\star)\big) - \big(R(x, y_1, g^\star) - R(x, y_0, g^\star)\big) \Big|, \qquad (A.32)$$

where $\kappa = 1/(1 + \exp(B))^2$. Therefore, we bound the left-hand side of (A.33) as

$$\mathcal{L}(R^\star, \bar{\mathcal{D}}) - \mathcal{L}(R, \bar{\mathcal{D}})$$

$$\leq -2\kappa^2 \cdot \mathbb{E}_{(x,y_1,y_0,g)\sim\mu_{\bar{\mathcal{D}}}} \left[ \Big| \big(R^\star(x, y_1, g) - R^\star(x, y_0, g)\big) - \big(R(x, y_1, g) - R(x, y_0, g)\big) \Big|^2 \right]$$

$$+ \frac{3}{N} \cdot \log \left( \frac{\mathcal{N}_\varepsilon(\mathcal{R}, \|\cdot\|_\infty)}{\delta} \right). \qquad (A.33)$$

Meanwhile, the reward difference term in (A.29), which is evaluated on responses sampled from $\pi^\star$ and $\pi_{\text{sft}}$, can be related to the reward difference evaluated on the data distribution $\mu_{\bar{\mathcal{D}}}$ via Assumption A.3 as follows,

$$\mathbb{E}_{x\sim d_0, y_1\sim\pi^\star(\cdot|x,g^\star), y_0\sim\pi_{\text{sft}}(\cdot|x,g^\star)} \left[ \big(R^\star(x, y_1, g^\star) - R^\star(x, y_0, g^\star)\big) - \big(R(x, y_1, g^\star) - R(x, y_0, g^\star)\big) \right]$$

$$\leq C_{\mu_{\bar{\mathcal{D}}}}(\mathcal{R}; \pi^\star, \pi_{\text{sft}}) \sqrt{\mathbb{E}_{(x,y_1,y_0,g)\sim\mu_{\bar{\mathcal{D}}}} \left[ \Big| \big(R^\star(x, y_1, g) - R^\star(x, y_0, g)\big) - \big(R(x, y_1, g) - R(x, y_0, g)\big) \Big|^2 \right]}.$$
$$(A.34)$$

Combining (A.33), (A.34), and (A.29) and defining

$$\Delta_R := \sqrt{\mathbb{E}_{(x,y_1,y_0,g)\sim\mu_{\bar{\mathcal{D}}}} \left[ \Big| \big(R^\star(x, y_1, g) - R^\star(x, y_0, g)\big) - \big(R(x, y_1, g) - R(x, y_0, g)\big) \Big|^2 \right]},$$
$$(A.35)$$

we obtain

$$\text{Gap}^{\pi^\star}(\widehat{\pi}) \leq \max_{R\in\mathcal{R}} \left\{ C_{\mu_{\bar{\mathcal{D}}}}(\mathcal{R}; \pi^\star, \pi_{\text{sft}}) \cdot \Delta_R - 2\eta^{-1}\kappa^2 \cdot \Delta_R^2 \right\} + \frac{3}{\eta N} \cdot \log \left( \frac{\mathcal{N}_\varepsilon(\mathcal{R}, \|\cdot\|_\infty)}{\delta} \right)$$

$$+ \beta \cdot \mathbb{E}_{x\sim d_0} \left[ \text{KL}\big(\pi^\star(\cdot|x,g^\star)\|\pi_{\text{ref}}(\cdot|x,g^\star)\big) - \text{KL}\big(\widehat{\pi}(\cdot|x,g^\star)\|\pi_{\text{ref}}(\cdot|x,g^\star)\big) \right]$$

$$\leq \frac{\big(C_{\mu_{\bar{\mathcal{D}}}}(\mathcal{R}; \pi^\star, \pi_{\text{sft}})\big)^2 \eta}{8\kappa^2} + \frac{3}{\eta N} \cdot \log \left( \frac{\mathcal{N}_\varepsilon(\mathcal{R}, \|\cdot\|_\infty)}{\delta} \right)$$

$$+ \beta \cdot \mathbb{E}_{x\sim d_0} \left[ \text{KL}\big(\pi^\star(\cdot|x,g^\star)\|\pi_{\text{ref}}(\cdot|x,g^\star)\big) \right], \qquad (A.36)$$

where the second inequality uses the fact that $az - bz^2 \leq a^2/(4b)$ for any $z \in \mathbb{R}$ and KL-divergence is non-negative. As a result, selecting $\varepsilon = (6 \cdot (1 + e^B) \cdot N)^{-1}$ and

$$\eta = 2\sqrt{6} \cdot \sqrt{\frac{\log\big(\mathcal{N}_\varepsilon(\mathcal{R}, \|\cdot\|_\infty)/\delta\big)}{N}}, \quad \beta = \frac{1}{\sqrt{N}}, \quad \kappa = \frac{1}{(1 + \exp(B))^2}, \qquad (A.37)$$

we prove that with probability at least $1 - \delta$ that

$$\text{Gap}^{\pi^\star}(\widehat{\pi}) \leq \sqrt{\frac{1}{N}} \cdot \left\{ \frac{\sqrt{6}}{4} \big(1 + \exp(B)\big)^2 \big( \big(C_{\mu_{\bar{\mathcal{D}}}}(\mathcal{R}; \pi^\star, \pi_{\text{sft}})\big)^2 + 1 \big) \iota \right.$$

$$\left. + \mathbb{E}_{x\sim d_0} \left[ \text{KL}\big(\pi^\star(\cdot|x,g^\star)\|\pi_{\text{ref}}(\cdot|x,g^\star)\big) \right] \right\}, \qquad (A.38)$$

where we denote $\iota = \sqrt{\log\big(\mathcal{N}_\varepsilon(\mathcal{R}, \|\cdot\|_\infty)/\delta\big)}$. Combining Theorem A.6, (A.20), and (A.38), we conclude the proof of Theorem A.4.

## A.6 TECHNICAL LEMMAS

**Lemma A.7** (Uniform concentration). Consider the negative log-likelihood loss in (A.2) and define the approximation error as $\varepsilon = (6 \cdot (1 + e^B) \cdot N)^{-1}$, where we assume that $R(x, y, g) \in [-B/2, B/2]$

for any $(R, x, y, g) \in \mathcal{R} \times \mathcal{X} \times \mathcal{Y} \times \mathcal{G}$. Suppose that the reward model class $\mathcal{R}$ has a finite $\varepsilon$-covering number $\mathcal{N}_\varepsilon(\mathcal{R}, \|\cdot\|_\infty) < \infty$. Then for any $\delta < 1/e$ it holds with probability at least $1 - \delta$ that

$$\mathcal{L}(R^\star, \bar{\mathcal{D}}) - \mathcal{L}(R, \bar{\mathcal{D}}) \tag{A.39}$$

$$\leq -2 \cdot \mathbb{E}_{(x,y_1,y_0,g)\sim\mu_{\bar{\mathcal{D}}}} \left[ D_{\text{Hellinger}}^2 \left( \mathbb{P}_{R^\star}(\cdot|x, y_1, y_0, g) \| \mathbb{P}_R(\cdot|x, y_1, y_0, g) \right) \right]$$

$$+ \frac{3}{N} \cdot \log \left( \frac{\mathcal{N}_\varepsilon(\mathcal{R}, \|\cdot\|_\infty)}{\delta} \right). \tag{A.40}$$

*Proof.* See the proof of Lemma D.1 in Liu et al. (2024), where we use the fact that $(x, g)$ follows a fixed distribution. $\square$

**Lemma A.8** (Difference of Sigmoid functions). For any real numbers $z_1, z_2 \in [-B/2, B/2]$, it holds that

$$\kappa \cdot |z_1 - z_2| \leq |\sigma(z_1) - \sigma(z_2)| \leq |z_1 - z_2|, \tag{A.41}$$

where the constant $\kappa = 1/(1 + \exp(B))^2$.

*Proof.* See the proof of Lemma D.2 in Liu et al. (2024). $\square$

# B EXPERIMENT DETAILS

## B.1 SETUP

We use the following prompt during training. Here, the reward values are the quality scores given by the judge models that exist in the preference dataset. The prompt is set as the system prompt whenever the LLM supports, such as Qwen2-7B-Instruct and Llama-3.1-8B-Instruct, and it is prefixed before the original prompt when the LLM doesn't support system prompting, such as Mistral-7B-Instruct-v0.3 and Gemma-2-9B-It.

```
Training prompt

You are an assistant that generates responses for the instruction
while implicitly achieving the following target score (on a scale of
1-10, where 1 is lowest and 10 is highest):
Overall score: {reward_value}.
```

At inference time, we use almost the same prompt, except that the goal score is the highest one, i.e., the overall score is 10.

```
Inference prompt

You are an assistant that generates responses for the instruction
while implicitly achieving the following target score (on a scale of
1-10, where 1 is lowest and 10 is highest):
Overall score: 10.
```

In our experiments using UltraFeedback, we directly leverage the LLM-as-Judge scores provided by GPT-4 in the dataset, which range from 1 to 10. For our method that is applied to on-policy data ranked by external reward models, including PairRM and ArmoRM, we apply linear transformations to normalize the resulting reward scores, ensuring they are scaled within the same 1 to 10 range.

For hyperparameters, we utilize a KL regularization coefficient of $\beta = 0.01$ in DPO, and we adopt the AdamW optimizer (Loshchilov, 2017). The batch size is set to $128$, with a learning rate of $5e-7$ and a warmup ratio of $0.1$. Furthermore, we observe that for models such as Qwen2-7B-Instruct and Gemma-2-9B-It on UltraFeedback, as well as Llama-3-8B-Instruct on on-policy data, both DPO and our proposed method yield improved performance when employing the conservative DPO (cDPO) technique (Mitchell, 2023). Consequently, for these models, we set the label smoothing hyperparameter from the Alignment Handbook (Tunstall et al., 2023a) to 0.3, while keeping it at 0 for the remaining models.

## B.2 FULL RESULTS

In Table 11, we present the full results on instruction-following benchmarks, which correspond to the performance illustrated in Figure 2 in the main text.

| | AlpacaEval 2.0 | | | MT-Bench | | | Arena-Hard-Auto | |
|---|---|---|---|---|---|---|---|---|
| | LC WR | WR | Avg. Len. | Avg. | 1st | 2nd | Score | Avg. Len. |
| Mistral-7B-Instruct-v0.3 | 19.65 | 15.40 | 1503 | 7.67 | 8.00 | 7.34 | 17.0 | 494 |
| +DPO (UltraFeedback) | 18.76 | 16.93 | 1643 | 7.66 | 7.92 | **7.40** | 17.6 | 504 |
| +DPO (Reward-Augmented) | **25.99** | **28.36** | 2270 | **7.69** | **8.02** | 7.36 | **18.3** | 883 |
| Qwen2-7B-Instruct | 20.93 | 18.22 | 1788 | 7.90 | 8.23 | 7.56 | 24.3 | 617 |
| +DPO (UltraFeedback) | 21.46 | 19.35 | 1797 | 8.33 | 8.72 | 7.93 | 21.9 | 553 |
| +DPO (Reward-Augmented) | **31.17** | **27.58** | 1789 | **8.47** | **8.93** | **7.97** | **30.1** | 644 |
| Llama-3.1-8B-Instruct | 24.79 | 27.38 | 2081 | 8.44 | 8.99 | 7.90 | 26.9 | 831 |
| +DPO (UltraFeedback) | 28.67 | 30.21 | 2053 | 8.47 | **9.01** | 7.93 | 33.0 | 1070 |
| +DPO (Reward-Augmented) | **31.20** | **35.93** | 2006 | 8.47 | 8.91 | **8.03** | **34.4** | 824 |
| Gemma-2-9B-It | 49.20 | 37.58 | 1572 | 8.54 | 8.81 | 8.28 | 42.8 | 541 |
| +DPO (UltraFeedback) | 50.70 | 35.02 | 1464 | 8.54 | 8.70 | **8.37** | 35.8 | 456 |
| +DPO (Reward-Augmented) | **59.27** | **54.56** | 1872 | **8.59** | **8.93** | 8.25 | **43.9** | 611 |
| SPPO | 55.60 | 49.61 | 1822 | 8.40 | 8.53 | 8.26 | 47.6 | 578 |
| +DPO (UltraFeedback) | 52.75 | 40.58 | 1544 | 8.41 | 8.78 | 8.04 | 40.4 | 457 |
| +DPO (Reward-Augmented) | **60.97** | **66.41** | 2543 | **8.73** | **9.06** | **8.41** | **49.0** | 761 |

Table 11: Results of the DPO models fine-tuned on UltraFeedback and on reward-augmented UltraFeedback. We evaluate on the instruction-following benchmarks including AlpacaEval 2.0, MT-Bench, and Arena-Hard-Auto.

We also provide the full comparison results with reward-augmented methods in Table 12.

| | Zephyr-SFT | DPO | DPA | SteerLM | NCA-P | NCA-R | INCA-P | INCA-R | Ours |
|---|---|---|---|---|---|---|---|---|---|
| LC Win Rate | 6.21 | 11.60 | 11.13 | - | 11.50 | 12.87 | 13.68 | 14.83 | 16.66 |
| Win Rate | 3.94 | 8.58 | 10.58 | 8.21 | 8.43 | 9.56 | 11.00 | 11.34 | 13.37 |
| Avg. Len. | 893 | 1240 | 1671 | 1585 | 1287 | 1364 | 1449 | 1338 | 1812 |

Table 12: Full comparison results with reward-augmented methods.

## B.3 MORE ABLATIONS

**Controllable Generation with Prompt.** In Table 13, we ablate how generations differ when changing the goal rewards in the system prompt. We observe that the AlpacaEval 2.0 scores of the Qwen2-7B-It+DPO (RA) model change accordingly as $g$ varies. However, using the same $g = 10$ prompt during inference for the Qwen2-7B-It+DPO (UF) model fails to give competitive results, indicating that our method is superior not only because of the additional system prompt.

| | $g = 10$ | $g = 8$ | $g = 6$ | UF ($g = 10$) |
|---|---|---|---|---|
| LC WR | **31.17** | 28.66 | 25.56 | 24.44 |
| WR | **27.58** | 25.57 | 18.88 | 20.75 |

Table 13: Performance when conditioned on different goal rewards in the inference prompt.

**Benefits of Learning from High-Quality Rejected Responses.** Using the UltraFeedback dataset, we construct two reward-augmented preference datasets by filtering out augmented data based on rejected responses with low and high reward values, respectively. Compared to our method, these datasets isolate the impact of excluding low- and high-reward rejected responses as goals. The evaluation results on AlpacaEval 2.0 are presented in Table 14. Learning from rejected high-reward samples demonstrates superior performance compared to the approach that excludes these samples.

| | Qwen2-7B-It | +DPO (UF) | +DPO (RA) | +DPO (RA filter high) | +DPO (RA filter low) |
|---|---|---|---|---|---|
| LC Win Rate | 20.93 | 21.46 | 31.17 | 29.36 | **31.81** |
| Win Rate | 18.22 | 19.35 | **27.58** | 27.04 | 27.28 |

Table 14: Ablation on the benefits of learning from high-quality rejected responses.

**Impact of the Reward Scale.** For the UltraFeedback dataset that contains response rewards in the range of 1-10, we relabel them to be in the range of 1-5 and 1-100 with linear transformation. Our method followed by DPO is then applied on these different scaled datasets. The results are shown in Table 15. It can be observed that our method is robust to the reward scales. Since our main experiments use the default 1-10 scale as in UltraFeedback, it is likely that the performance can be further boosted, e.g., by adopting the 1-100 scale.

|             | Qwen2-7B-It | +DPO (UF) | +DPO (RA, 5) | +DPO (RA 10) | +DPO (RA 100) |
|-------------|-------------|-----------|--------------|--------------|---------------|
| LC Win Rate | 20.93       | 21.46     | 29.85        | 31.17        | **31.81**     |
| Win Rate    | 18.22       | 19.35     | 26.12        | 27.58        | **27.96**     |

Table 15: Ablation on the impact of the reward scale demonstrates the robustness of our method.

