# OpenReview forum: "Reward-Augmented Data Enhances Direct Preference Alignment of LLMs"
_ICLR.cc/2025/Conference — Submitted to ICLR 2025_

### Official Review · Reviewer_Jv4d · 2024-10-21

**Soundness:** 3
**Presentation:** 2
**Contribution:** 2
**Rating:** 5
**Confidence:** 4

**Summary:**

This paper proposes a method for learning from a reward-augmented preference dataset. The method is straightforward. During training, the LLM is prompted with a special text like "Generate a response of reward 5". If such a prompt is given, only the response that has reward 5 is preferred during DPO training, the other response (even it has a higher reward) is considered as dispreferred one. This allows us not only to learn from preferred response data but also learn from the suboptimal response. This also allows the leverage of quantitative reward values. Exps is conducted on reward datasets like UltraFeedback, and show improvement over the DPO method.

**Strengths:**

1. The proposed method is straightforward and easy to understand.
2. Experiments show that the proposed method outperforms DPO on multiple base models and various benchmarks.
3. The algorithm can somewhat mitigate the unlearning problem of DPO, which might be useful.

**Weaknesses:**

1. **Conditioning on low reward data is only empirically useful.** Despite improvements indicated by the experiments, I find the method to be highly empirical and lacks theoretical guarantees. During sampling, the LLM is always prompted to generate only high-reward responses, which makes learning low-reward prompts during training useful only because it proves "generalization".

2. **Lack of comparison with other reward-augmented methods**. One main claimed contribution of the paper is being able to learn from quantitative rewards, but it did not compare with any reward-augmented methods. Related prior works are

Noise Contrastive Alignment of Language Models with Explicit Rewards   https://arxiv.org/abs/2402.05369

Ulma: Unified language model alignment with demonstration and point-wise human preference https://arxiv.org/abs/2312.02554

3. **Reward labels can be extremely noisy.**  In the experiments, all datasets use GPT-4 generated reward, which is much more deterministic than human reward. Preference learning exactly tries to avoid noises in human evaluations and only considers preference order. If the dataset reward is given by a human instead of GPT4. One response can be of multiple rewards according to different annotators. So it is unreasonable when you prompt the LLM to generate a response of reward 9, the response of reward 10 is "dispreferred".

4. **Reward values are continuous.** The method conditioned the LLM on a specific given reward value. However, since the reward signal is an inherently continuous value, it is not very suitable for use as the condition. "Generate a response of reward 3.75" does not make much sense, compared with "Generate a response that is truthful/contains no mathematic mistakes (some deterministic requirement)"

**Questions:**

Above

---

> ### Author Response · Authors · 2024-11-27
>
> We thank the reviewer for identifying our work's technical contributions. The valuable comments have helped us improve our manuscript (marked **blue** in the revision). Below are our specific responses to the questions raised by the reviewer:
>
> **Weakness 1: Conditioning on low reward data is only empirically useful.  Despite improvements indicated by the experiments, I find the method to be highly empirical and lacks theoretical guarantees. During sampling, the LLM is always prompted to generate only high-reward responses, which makes learning low-reward prompts during training useful only because it proves "generalization".**
>
> - Direct alignment on our augmented reward-relabeled data ensures that the high-quality rejected responses are not suppressed, since the LLM is trained to generate responses corresponding to different reward values, enabling it to become aware of and adapt to these reward distinctions.
> - We also added theoretical results at the end of **Section 4 and in Appendix A**. Specifically, we showed that our method attains global convergence to the optimal policy and the suboptimality decays at the order of $N^{-1/2}$ ($N$ denotes the size of the reward-augmented preference dataset), which provides a theoretical justification for the strong empirical performance of the reward-augmented DPO introduced in this paper. Unlike prior works on goal-conditioned RL with supervised learning [4, 5], which typically establish weaker results such as local performance improvements or the optimization of a lower bound on $J(\pi)$, our analysis guarantees global convergence to the optimal policy. This distinction underscores the significance of integrating DPO-like methods with goal-conditioned approaches.
>
> **Weakness 2: Lack of comparison with other reward-augmented methods. One main claimed contribution of the paper is being able to learn from quantitative rewards, but it did not compare with any reward-augmented methods. Related prior works are [1] and [2].**
> We compared with reward-conditioned baselines in experiments including DPA [3], SteerLM [4], and RPO [7], and we thank the reviewer for pointing out the great missing related work [1, 2]. We have made two modifications in the revision:
> - We added the following paragraph to the related work section:
> "Notably, Noise Contrastive Alignment [1] and Unified Language Model Alignment [2] introduce unified frameworks for alignment with binarized or reward datasets by leveraging (information) noise contrastive estimation and a hybrid of SFT with point-wise DPO, respectively. In contrast, our work focuses on addressing the limitations of direct alignment algorithms with data relabeling (on implicit-reward augmented binarized or reward datasets), and do not make algorithm changes."
> - Experimentally, we add comparsions with Noise Contrastive Alignment [1]. Specifically, we compare the Zephyr-SFT models fine-tuned with our method and DPO, DPA [3], SteerLM [4], NCA-preference, NCA-reward, InfoNCA-preference, InfoNCA-reward, all on UltraFeedback. The results on the AlpacaEval 2.0 benchmark are shown as follows:
>
> |                   | LC Win Rate | Win Rate  | Avg. Len. |
> |-------------------|-------------|-----------|-----------|
> | Zephyr-SFT        | 6.21        | 3.94      | 893       |
> | DPO        | 11.60       | 8.58      | 1240      |
> | DPA       | 11.13       | 10.58     | 1671      |
> | SteerLM    | -    | 8.21   | 1585      |
> | NCA-preference       | 11.50       | 8.43      | 1287      |
> | NCA-reward       | 12.87       | 9.56      | 1364      |
> | InfoNCA-preference       | 13.68       | 11.00      | 1449      |
> | InfoNCA-reward       | 14.83       | 11.34      | 1338      |
> | Ours | **16.66**   | **13.37** | 1812      |
>
> It can be observed that NCA and InfoNCA are strong baselines and significantly outperform SteerLM, DPO, and DPA. Besides, our method improves DPO by a considerable margin. We have incorporated the above results in our revised manuscript (**Figure 4 and Appendix B.2**).

---

> > ### Author Response · Authors · 2024-11-27
> >
> > **Weakness 3: Reward labels can be extremely noisy. In the experiments, all datasets use GPT-4 generated reward, which is much more deterministic than human reward. Preference learning exactly tries to avoid noises in human evaluations and only considers preference order. If the dataset reward is given by a human instead of GPT4. One response can be of multiple rewards according to different annotators. So it is unreasonable when you prompt the LLM to generate a response of reward 9, the response of reward 10 is "dispreferred".**
> >
> > - For two responses that are of similar qualities, such as the pairs that are assigned scores of $9$ and $10$ by judge models or the former response is preferred by slightly more annotators, direct alignment algorithms such as DPO may strive to maximize the reparameterized reward gap between them. This will cause unnecessary unlearning of the high-quality $9$-scored rejected response, potentially diminishing the model’s performance by discarding valuable alternatives. On the contrary, our method extracts their common features as desired behaviors.
> > - The inherent noise in the reward scores (e.g., the $9$-scored response will actually be preferred by more human annotators and should be scored $10$, and the $10$-scored response will be scored $9$) has minimal impact on our method since the resulting model learns from both responses and the common pattern between these high-quality responses is extracted. However, DPO still tries to unlearn one of these high-quality responses, depending on which one is rejected in the dataset.
> >
> > **Weakness 4: Reward values are continuous. The method conditioned the LLM on a specific given reward value. However, since the reward signal is an inherently continuous value, it is not very suitable for use as the condition. "Generate a response of reward 3.75" does not make much sense, compared with "Generate a response that is truthful/contains no mathematic mistakes (some deterministic requirement)"**
> >
> > - Conditioning the LLM on reward values using prompt has been shown effective in various prior works [3, 4, 5, 6] that perform controllable optimization of LLMs.
> > - As suggested by the reviewer, we conduct additional experiments by relabelling with deterministic coarse-grained goals. Specifically, in the system prompt of Qwen2-7B-Instruct, instead of using the integer reward values from UltraFeedback, we use coarse-grained goals such as "You are a helpful/unhelpful assistant". The results on the AlpacaEval 2.0 benchmark are shown as follows. It can be observed that relabelling with coarse-grained goals improves the base model and DPO, but its performance still lags behind the fine-grained reward values considered in the main experiments. That being said, it is likely that our prompt for the coarse-grained goals is not optimal, and we leave finding more optimal prompts as future works.
> >
> > |                   | LC Win Rate | Win Rate  | Avg. Len. |
> > |-------------------|-------------|-----------|-----------|
> > | Qwen2-7B-Instruct        | 20.93       | 18.22     | 1788       |
> > | +DPO        | 21.46       | 19.35      | 1240      |
> > | +Ours (coarse) | 25.05 | 22.72 | 1825      |
> > | +Ours (fine) | **31.17**  | **27.58** | 1789   |
> >
> > ---
> > We hope the reviewer could consider raising the score if we resolved the reviewer's concerns. We would be happy to have further discussions if the reviewer has any additional questions or comments.
> >
> > ---
> > [1] Chen et al. ''Noise Contrastive Alignment of Language Models with Explicit Rewards.''\
> > [2] Cai et al. "ULMA: Unified Language Model Alignment with Human Demonstration and Point-wise Preference."\
> > [3] Wang et al. "Arithmetic Control of LLMs for Diverse User Preferences: Directional Preference Alignment with Multi-Objective Rewards."\
> > [4] Dong et al. "SteerLM: Attribute Conditioned SFT as an (User-Steerable) Alternative to RLHF."\
> > [5] Wang et al. "Conditional Language Policy: A General Framework for Steerable Multi-Objective Finetuning."\
> > [6] Guo et al. "Controllable Preference Optimization: Toward Controllable Multi-Objective Alignment."\
> > [7] Nvidia "Nemotron-4 340B Technical Report."

---

> > > ### Comment · Reviewer_Jv4d · 2024-11-27
> > >
> > > I appreciate the reviewer for providing the detailed rebuttal response. My concerns for **W2** and **W4** are mostly addressed. I thus raise my score to 5 and might further update my rating based and discussions and opinions of other reviewers.

---

> > > > ### Author Response · Authors · 2024-11-27
> > > >
> > > > We thank the reviewer for reviewing our rebuttal and raising the score! We are delighted to know that our response addressed your concerns regarding W2 and W4. Could you please clarify if there are any remaining issues with W1 and W3, and how we might address them? We would be more than happy to have further discussions if the reviewer has any additional questions or comments.

---

> > > > > ### Author Response · Authors · 2024-11-29
> > > > >
> > > > > We thank the reviewer for the valuable insights and comments, which have significantly enhanced the quality of our manuscript. We are glad that our previous responses addressed your concerns about **Weakness 2** and **Weakness 4**.
> > > > > In our rebuttal response, we also incorporated theoretical results to better motivate and support our method to address your **Weakness 1** about the lack of theoretical guarantees. Besides, to address your **Weakness 3**, we described the limitations of DPO and how our method addresses them using the example you provided.
> > > > > We hope you could consider raising the score if we resolved your concerns and would like to have further discussions if you have any additional questions or comments.

---

> > > > > > ### Author Response · Authors · 2024-12-01
> > > > > >
> > > > > > Dear Reviewer,
> > > > > >
> > > > > > Since the discussion period ends tomorrow, we would like to follow up to see if our responses addressed the reviewer’s concerns. We would be happy to have further discussions if the reviewer has any additional questions or comments.
> > > > > >
> > > > > > Best,
> > > > > > Authors of Submission 8745

---

### Official Review · Reviewer_LQmQ · 2024-11-04

**Soundness:** 3
**Presentation:** 3
**Contribution:** 2
**Rating:** 3
**Confidence:** 4

**Summary:**

This paper tackles the limitations of direct alignment in LLMs by introducing reward-conditioned policies. The authors propose a data relabeling method that incorporates reward goals into preference datasets, allowing the model to learn from both high- and low-quality responses. By conditioning on the highest reward at inference, the approach enhances LLM performance across benchmarks, improving alignment and reducing overfitting.

**Strengths:**

This paper presents a novel approach to preference alignment in LLMs by introducing reward-conditioned policies. The method enhances model performance by conditioning on reward values, enabling LLMs to better generalize to high-quality responses and mitigate the unlearning Issue.

**Weaknesses:**

1. The method relies heavily on reward labels, whether in directly constructing the reward-augmented dataset or through a multi-attribute rewards dataset. The actual impact of these rewards on data quality remains unexplored. It is unclear if the model would benefit more from high-reward rejected answers than from low-reward chosen answers, which warrants further investigation.
2. Learning from preference usually involves guiding the model towards specific output styles, such as aligning to certain type of instructions or adapting to varied language tones. The authors used previously unseen instructions in DPO training that match existing data, potentially limiting the model's adaptability. The experimental results require additional examples and detailed explanations to clarify this impact. And in appendix $D_N^i$ training prompts should be added.
3. The authors’ setup inadequately considers response scores (rewards), despite splitting the augmented dataset into two categories, $r_w^i$ and $r_l^i$. The rationale categorization of additional quality dimensions should further considered beyond binary labels.
4. The proposed approach and insights lack a strong foundation. Restructuring the dataset (new pairs) based on reward intuitively seems more effective. In contrast, positive and negative goal-conditioned learning does not necessarily ensure that higher-quality responses receive more attention.
5. The formatting in line 200 contains typographical errors. $r_{g = r_l^i}(x, y_w^i) = 0$.

**Questions:**

1. How does the reliance on reward labels impact the overall data quality, and would the model benefit more from incorporating high-reward rejected responses compared to low-reward chosen ones?
2. How does the use of previously unseen instructions in DPO training affect the model's adaptability, and could additional examples or detailed explanations enhance the clarity of the experimental results?
3. Could the authors further justify the categorization of responses beyond binary labels, and would explain how positive and negative goal-conditioned learning enhances attention of data with varying quality.

---

> ### Author Response · Authors · 2024-11-27
>
> We thank the reviewer for identifying our work's novelty, soundness, and technical contributions. The valuable comments have helped us improve our manuscript (marked **blue** in the revision). Below are our specific responses to the questions raised by the reviewer:
>
> **Weakness 1: The method relies heavily on reward labels, whether in directly constructing the reward-augmented dataset or through a multi-attribute rewards dataset. The actual impact of these rewards on data quality remains unexplored. It is unclear if the model would benefit more from high-reward rejected answers than from low-reward chosen answers, which warrants further investigation.**
> Following the reviewer's suggestions, we conduct additional experiments to show the benefits of learning from the rejected high-reward samples. Specifically, using the UltraFeedback dataset, we construct two reward-augmented preference datasets by filtering out augmented data based on rejected responses with low and high reward values, respectively, which we denote as **Ours-filter-low** and **Ours-filter-high**. Compared to our method, these datasets isolate the impact of excluding low- and high-reward rejected responses as goals. The results on AlpacaEval 2.0 are shown as follows:
>
> |                   | LC Win Rate | Win Rate  |
> |-------------------|-------------|-----------|
> | Qwen2-7B-Instruct | 20.93        | 18.22      |
> | +DPO        | 21.46        | 19.35     |
> | +Ours        | 31.17        | **27.58**      |
> | +Ours-filter-high  | 29.36   | 27.04 |
> | +Ours-filter-low  | **31.81**   | 27.28 |
>
> It can be observed that learning from rejected high-reward samples   (**Ours** and **Ours-filter-low**) demonstrates superior performance compared to the approach that excludes these samples (**Ours-filter-high**), demonstrating the benefits of keeping the rejected high-reward samples.
>
> **Weakness 2: Learning from preference usually involves guiding the model towards specific output styles, such as aligning to certain type of instructions or adapting to varied language tones. The authors used previously unseen instructions in DPO training that match existing data, potentially limiting the model's adaptability. The experimental results require additional examples and detailed explanations to clarify this impact. And in appendix $\mathcal{D}_N^i$ training prompts should be added.**
> - In RLHF or RLAIF, it is common practice to use instructions that are different from the ones in SFT or in previous alignment (e.g., DPO) iterations. Since our method does not alter the direct alignment algorithms, it is likely that it will share similarities with DPO, such as guiding the model towards specific output styles to align with human or AI preferences.
> - Following the reviewer's suggestion, we report the performance of the DPO baseline and our method fine-tuned from the Zephyr-SFT model, where a large portion of instructions during alignment also appear in the SFT stage. The results on the AlpacaEval 2.0 benchmark are as follows. It can be observed that applying DPO on both the original preference dataset and our reward-augmented dataset improves the performance with longer output length, which might be a sign of learning generation styles preferred by judge models.
>
> |                   | LC Win Rate | Win Rate  | Avg. Len. |
> |-------------------|-------------|-----------|-----------|
> | Zephyr-SFT        | 6.21        | 3.94      | 893       |
> | +DPO        | 11.60       | 8.58      | 1240      |
> | +Ours | **16.66**   | **13.37** | 1812      |
>
> - In fact, it is widely observed that alignment can considerably improve scores judged by LLMs or humans but can decrease the reasoning abilities of the LLM, which is known as the alignment tax. In our experiments (Table 3), we observed that our method significantly and consistently improves the base models on instruction-following benchmarks that use LLM as a judge, and moderately improves most of the academic benchmarks. This indicates that our method does not suffer from severe alignment tax.

---

> > ### Author Response · Authors · 2024-11-27
> >
> > **Weakness 3: The authors’ setup inadequately considers response scores (rewards), despite splitting the augmented dataset into two categories, $r_w^i$ and $r_l^i$. The rationale categorization of additional quality dimensions should be further considered beyond binary labels.**
> >
> > - We augment the dataset by conditioning on $r^w_i$ and $r^l_i$ since preference datasets during RLAIF are constructed as $\mathcal{D}=[x^i, y_w^i, y_l^i, r_w^i, r_l^i]_{i=1}^N$ and for the goal-conditioned reward $R(x, y, G) = - (g-r(x,y))^2$, we only know it is non-negative when $g=r_w^i, y=y_w^i$ and $g=r_l^i, y=y_l^i$. This leads to the two categories appearing in our method.
> > - We would like to emphasize that the the response scores can be of various forms from single- to multi-dimensional. In experiments (Table 10), we ablated using multi-attribute rewards that correspond to quality dimensions beyond overall qualities (e.g., complexity, honesty, and helpfulness) and found that the resulting models achieve SOTA performance on AlpacaEval 2.0:
> >
> > |                   | LC Win Rate | Win Rate  |
> > |-------------------|-------------|-----------|
> > | Llama-3-8B-Instruct        | 22.92    | 23.15      |
> > | +DPO        | 42.32       | 48.73      |
> > | +DPO (Ours) | **56.57**   | **52.19** |
> >
> > **Weakness 4: The proposed approach and insights lack a strong foundation. Restructuring the dataset (new pairs) based on reward intuitively seems more effective. In contrast, positive and negative goal-conditioned learning does not necessarily ensure that higher-quality responses receive more attention.**
> >
> > - Direct alignment on our augmented reward-relabeled data ensures that the high-quality rejected responses are not suppressed, since the LLM is trained to generate responses corresponding to different reward values, enabling it to become aware of and adapt to these reward distinctions.
> > - We also added theoretical results at the end of **Section 4 and in Appendix A**. Specifically, we showed that our method attains global convergence to the optimal policy and the suboptimality decays at the order of $N^{-1/2}$ ($N$ denotes the size of the reward-augmented preference dataset), which provides a theoretical justification for the strong empirical performance of the reward-augmented DPO introduced in this paper. Unlike prior works on goal-conditioned RL with supervised learning [1, 2], which typically establish weaker results such as local performance improvements or the optimization of a lower bound on $J(\pi)$, our analysis guarantees global convergence to the optimal policy. This distinction underscores the significance of integrating DPO-like methods with goal-conditioned approaches.
> >
> > **Weakness 5: The formatting in line 200 contains typographical errors. $r_{g=r_l^i}(x, y_w^i) = 0$**
> >
> > We thank the reviewer for pointing this out. The typo has been corrected in the revision.
> >
> > **Question 1: How does the reliance on reward labels impact the overall data quality, and would the model benefit more from incorporating high-reward rejected responses compared to low-reward chosen ones?**
> >
> > We have addressed the questions with additional ablation studies. Please see our response to your **Weakness 1**.
> >
> > **Question 2: How does the use of previously unseen instructions in DPO training affect the model's adaptability, and could additional examples or detailed explanations enhance the clarity of the experimental results?**
> >
> > We have addressed the questions with additional examples and detailed explanations. Please see our response to your **Weakness 2**. We thank the reviewer for these valuable discussions, which we have incorporated into our experiments (**Section 6.2 and Appendix B.2**) in the revision to enhance clarity.
> >
> > **Question 3: Could the authors further justify the categorization of responses beyond binary labels, and would explain how positive and negative goal-conditioned learning enhances attention of data with varying quality.**
> >
> > We have addressed the questions by justifying the usage of binary labels and multi-dimensional quality scores in our response to your **Weakness 3** and adding additional ablation studies about varying-quality data in our response to your **Weakness 1**. We thank the reviewer for these valuable discussions, which we have incorporated into the revised manuscript (**Appendix B.3**).
> >
> >
> > ---
> > We hope the reviewer could consider raising the score if we resolved the reviewer's concerns. We would be happy to have further discussions if the reviewer has any additional questions or comments.
> >
> > ---
> > [1] Yang et al. "Rethinking goal-conditioned supervised learning and its connection to offline rl."\
> > [2] Ghosh et al. "Learning to reach goals via iterated supervised learning."

---

> > > ### Author Response · Authors · 2024-11-28
> > >
> > > Dear Reviewer,
> > >
> > > Since the discussion period ends in **5** days, we would like to follow up to see if our rebuttal responses addressed the reviewer’s concerns. We would be more than happy to have further discussions if the reviewer has any additional questions or comments.
> > >
> > > Best,\
> > > Authors of Submission 8745

---

> > > > ### Author Response · Authors · 2024-11-29
> > > >
> > > > We thank the reviewer for the valuable insights and comments, which have significantly enhanced the quality of our manuscript. In our previous response, we added ablation studies by filtering different-quality responses as you suggested to address your **Weakness 1** and **Question 1** about the data quality. To resolve your **Weakness 2** and **Question 2** about the impact of DPO on the model adaptability, we added more results on applying DPO and our method on SFT models, and explained our experimental results such as the alignment tax. Besides, we clarified the design of two categories and reported the performance of our method using multi-attribute rewards to address your **Weakness 3** and **Question 3**. Furthermore, we incorporated theoretical results to better motivate and support our method to address your **Weakness 4** about "lack a strong foundation", and corrected the typo to resolve your **Weakness 5**. We hope you could consider raising the score if we resolved your concerns and would like to have further discussions if you have any additional questions or comments.

---

> > > > > ### Author Response · Authors · 2024-12-01
> > > > >
> > > > > Dear Reviewer,
> > > > >
> > > > > Since the discussion period ends tomorrow, we would like to follow up to see if our responses addressed the reviewer’s concerns. We would be happy to have further discussions if the reviewer has any additional questions or comments.
> > > > >
> > > > > Best,
> > > > > Authors of Submission 8745

---

### Official Review · Reviewer_Nw7q · 2024-11-08

**Soundness:** 3
**Presentation:** 3
**Contribution:** 2
**Rating:** 6
**Confidence:** 3

**Summary:**

This paper addresses fundamental limitations in direct preference alignment algorithms that use binary preference data, where models tend to unnecessarily unlearn high-quality rejected responses while indiscriminately learning low-quality chosen ones. The authors propose a simple reward-augmented data relabeling method that conditions the training data on explicit quality scores, which allows models to learn patterns across the full quality spectrum. This approach is orthogonal to existing direct alignment algorithms and can be applied with any direct preference algorithm.

**Strengths:**

1. The method is straightforward to implement, requires no architectural changes, and can be easily integrated with existing alignment algorithms.

2. The method is versatile since it works with both offline and online preference data and can handle multi-dimensional reward attributes.

**Weaknesses:**

1.  While the paper uses a 1-10 scale, there's no analysis of how different scoring scales (e.g., 1-5, 1-100) might affect performance or what the optimal scale might be.

2. The paper does not include human evaluation studies to validate the effectiveness of reward-augmented responses.

3. The paper would benefit from a discussion of limitations and potential future directions.

**Questions:**

What led to choosing the 1-10 scale for quality scores, and have you explored whether other scoring ranges might be more effective for reward conditioning?

---

> ### Author Response · Authors · 2024-11-27
>
> We thank the reviewer for identifying our work's novelty, soundness, and technical contributions. The valuable comments have helped us improve our manuscript (marked **blue** in the revision). Below are our specific responses to the questions raised by the reviewer:
>
> **Weakness 1: While the paper uses a 1-10 scale, there's no analysis of how different scoring scales (e.g., 1-5, 1-100) might affect performance or what the optimal scale might be.**
> We follow the reviewer's suggestions to conduct ablations on the impact of reward scales. Specifically, for the UltraFeedback dataset that contains response rewards in the range of 1-10, we relabel them to be in the range of 1-5 and 1-100 with linear transformation. Our method is then applied to these different scaled datasets. The results are shown in the following table:
> |                   | LC Win Rate | Win Rate  |
> |-------------------|-------------|-----------|
> | Qwen2-7B-Instruct        | 20.93       | 18.22    |
> | +DPO        | 21.46       | 19.35      |
> | +Ours (5 scale ) | 29.85  | 26.12 |
> | +Ours (10 scale ) | 31.17  | 27.58 |
> | +Ours (100 scale ) | **31.81**  | **27.96** |
>
> It can be observed that our method is robust to the reward scales. Since our paper uses the default 1-10 scale as in UltraFeedback, it is likely that the performance can be further boosted, such as the 1-100 scale. The above results and discussion have been added to **Appendix B.3** in the revision.
>
> **Weakness 2: The paper does not include human evaluation studies to validate the effectiveness of reward-augmented responses.**
>
> Our work mainly studied RL from AI feedback, and in experiments, we follow previous works and test our models on various benchmarks, including instruction-following tasks that use LLM as a judge, as well as academic benchmarks. We are not aware of benchmarks that use human evaluations and are cheap to run. If the reviewer has any such benchmarks in mind, please let us know and we will include the results in our manuscript.
>
> **Weakness 3: The paper would benefit from a discussion of limitations and potential future directions.**
>
> We thank the reviewer for the suggestion. The following paragraph has been added to the last section of our revision:
> "Although the proposed method consistently shows considerable improvements on instruction-following benchmarks, the improvements on academic benchmarks are marginal, which are also observed in previous direct alignment algorithms. Besides, instruction-following benchmarks adopt LLM-as-a-judge and have the risk of favoring specific output styles. Human evaluations might be needed to determine the benefit of our method. Since our method is compatible with any direct alignment algorithm, future works also include applying our method to other algorithms beyond DPO."
>
> **Question 1: What led to choosing the 1-10 scale for quality scores, and have you explored whether other scoring ranges might be more effective for reward conditioning?**
> - The 1-10 scale is chosen because our main experiments use the UltraFeedback dataset, which has a default scale of 1-10 for the quality scores.
> - As suggested by the reviewer, we have ablated other scoring ranges, including 1-5 and 1-100. We observed that the method is robust to the scoring ranges and it is likely that the 1-10 range is not the optimal one. Please see our responses to your **Weakness 1** and **Appendix B.3** in the revision for details.
>
> ---
> We hope the reviewer could consider raising the score if we resolved the reviewer's concerns. We would be happy to have further discussions if the reviewer has any additional questions or comments.

---

> > ### Author Response · Authors · 2024-11-28
> >
> > Dear Reviewer,
> >
> > Since the discussion period ends in **5** days, we would like to follow up to see if our rebuttal responses addressed the reviewer’s concerns. We would be more than happy to have further discussions if the reviewer has any additional questions or comments.
> >
> > Best,\
> > Authors of Submission 8745

---

> > > ### Author Response · Authors · 2024-11-29
> > >
> > > We thank the reviewer for the valuable insights and comments, which have significantly enhanced the quality of our manuscript. In our previous response, we added ablation studies on the reward scale as you suggested to address your **Weakness 1** and **Question 1**. Besides, we clarified how and why we designed our experiments to address your **Weakness 2** about human evaluations, and added a paragraph of limitations and future directions to address your **Weakness 3**. We hope you could consider raising the score if we resolved your concerns and would like to have further discussions if you have any additional questions or comments.

---

> > > > ### Comment · Reviewer_Nw7q · 2024-12-03
> > > >
> > > > Thank you for your detailed responses and revisions to address the concerns raised. While I appreciate the effort to improve the manuscript, I will maintain my current scoring due to a critical limitation: The benchmarks rely solely on LLM-as-a-judge evaluation, which risks favoring certain output styles. The lack of human evaluation makes it challenging to assess the method's real-world utility and robustness. This remains a significant limitation. I encourage strengthening the work through additional evaluation approaches. Thank you for addressing the other points comprehensively.

---

> > > > > ### Author Response · Authors · 2024-12-03
> > > > >
> > > > > We thank the reviewer for the valuable comments and feedback. We are happy to know that most of your concerns are addressed. We address your remaining concerns regarding human evaluations as follows:
> > > > > - Our method does **not** make algorithm changes, so the concern of overfitting certain output styles (if exists) applies to all (direct) alignment algorithms.
> > > > > - The benchmarks we adopted are **not** solely based on LLM-as-a-judge. To address the concerns that RLHF risks favoring certain output styles, we have conducted evaluations on various academic multi-choice QA benchmarks, including GSM8K, GPQA, MUSR, TruthfulQA, BBH, and ARC (Table 3).
> > > > > - The results show that our method increases the performance on most of the benchmarks, indicating that **style-overfitting (or alignment tax) is not a severe issue here**. Otherwise, the model will not follow the required question-answering format as observed in some RLHF papers [1, 2].
> > > > > - In fact, DPO is not necessarily overfitting to output styles favored by humans or AI, depending on the construction of the data. For example, adding more math and coding preference pairs makes the LLM better at reasoning [3].
> > > > > - The benchmarks and setups in our experiments follow from many previous works in alignment. While we acknowledge the value of human evaluations, they are costly and are not standard practice in most RLHF and alignment papers.
> > > > >
> > > > > Did our responses address your concerns?
> > > > >
> > > > > ---
> > > > >
> > > > > [1] Pal et al. "Smaug: Fixing Failure Modes of Preference Optimisation with DPO-Positive."\
> > > > > [2] Meng et al. "SimPO: Simple Preference Optimization with a Reference-Free Reward."\
> > > > > [3] Dong et al. "RLHF Workflow: From Reward Modeling to Online RLHF."

---

### Official Review · Reviewer_ofzy · 2024-11-08

**Soundness:** 2
**Presentation:** 2
**Contribution:** 3
**Rating:** 6
**Confidence:** 4

**Summary:**

This paper proposes augmenting reward modeling datasets with goal-conditioning. In this setting, one requests a response that has a particular reward score and assigns a binary label as to whether or not the datapoint achieves that score. Extensive experiments show that doing so improves the win rate and other capability-related evaluations of the model over relevant baselines.

**Strengths:**

The empirical results in Section 6 are comprehensive across different capability and alignment evaluations as well as different models. The trends in the results are fairly consistent across settings, I think. The method itself is computationally efficient and simple to implement.

**Weaknesses:**

1. I am confused about Tables 1 and 2. How did you compute $\pi^*$? What are the assumptions used? If you assume a standard setting like the DPO paper, you would never get a policy that always generates one response over the other (because Bradley-Terry inherently assumes some noise). Also, just more realistically, LLMs are softmax parametrized, and the probability of a response can never be exactly 0. Overall, I am confused about Section 3.1, where, for example, "reward sparsity" is used to describe something different than what it traditionally means (sparsity over trajectory time, not over the dataset) and the logic is unclear. This makes the motivation for the algorithm unclear -- Section 3.2 is titled way too strongly for a method that has no clear theoretical backing. This weakness is the reason for giving a 6 instead of an 8. It is hard to justify accepting a paper with such vague and possibly incorrect text, even if the idea and method are performant.

2. I don't understand the method. All datapoints have only a binary indicator that one response was preferred over the other. So how are you determining the value of $r_w^i$ and $r_l^i$? Without some additional information, they would all be the same.

3. I think the accuracy of the reward model matters a lot, and PairRM and ArmoRM are both pretty good reward models from my understanding. It might be better to do a more stark separation where eg one flips the ArmoRM score and sees what happens. I think this would be interesting, since LMs usually don't use the context in the way that we expect them to.

Minor: I would not use "unlearning" as a term to describe the phenomenon since that has a very different meaning in ML literature.

**Questions:**

See above

---

> ### Author Response · Authors · 2024-11-27
>
> We thank the reviewer for identifying our work's novelty, soundness, and technical contributions. The valuable comments have helped us improve our manuscript (marked **blue** in the revision). Below are our specific responses to the questions raised by the reviewer:
>
>
> **Weakness 1.1:  I am confused about Tables 1 and 2. How did you compute  $π^*$? What are the assumptions used? If you assume a standard setting like the DPO paper, you would never get a policy that always generates one response over the other (because Bradley-Terry inherently assumes some noise). Also, just more realistically, LLMs are softmax parametrized, and the probability of a response can never be exactly 0.**
>
> - $\pi^*$ is defined as the policy that maximizes the reward learned from the preference dataset $\mathcal{D}$, i.e., $\pi^*(y\mid x) = argmax_\pi E_{x\sim\mathcal{D}, y\sim\pi(\cdot\mid x)}[r(x, y)]$. We may follow standard RLHF and assume that the true preference distribution can be modeled by Bradley-Terry, i.e., $r(x, y) = argmax_r E_{(x,y_w,y_l)\sim\mathcal{D}}[\log\sigma(r(x, y_w) - r(x, y_l))]$, where $\sigma(\cdot)$ is the logistic function.
> - As an example, in Table 1, $\mathcal{D}$ contains a single preference pair $y_1 > y_2$.  The fitted reward thus assigns a higher value of $y_1$ than $y_2$, i.e., $r(x, y_1) > r(x, y_2)$. To maximize $r$, the optimal policy is $\pi^*(y_1\mid x) = 1$.
> - Since we studied tabular settings in Section 3, for clarity, we considered nonparametric policies without KL regularization. The reviewer is correct that in the KL-regularized formula in DPO or when the policy is softmax parameterized, $\pi^*(y_1\mid x) < 1$. Nevertheless, even if we follow the exact DPO formulation by defining the reference policy as a uniform distribution $\pi_{\text{ref}}(y_1\mid x)=\pi_{\text{ref}}(y_2\mid x)=0.5$, $\pi^*(y_1\mid x)$ is still arbitrarily larger than $\pi^*(y_2\mid x)$.
>
> **Weakness 1.2: Overall, I am confused about Section 3.1, where, for example, "reward sparsity" is used to describe something different than what it traditionally means (sparsity over trajectory time, not over the dataset) and the logic is unclear. This makes the motivation for the algorithm unclear -- Section 3.2 is titled way too strongly for a method that has no clear theoretical backing. This weakness is the reason for giving a 6 instead of an 8. It is hard to justify accepting a paper with such vague and possibly incorrect text, even if the idea and method are performant.**
>
> - The reward sparsity in Section 3.1 refers to the sparsity of the highest-reward data. The goal-conditioned reward (defined in Sec. 4) $R(x, y, g=r_{\text{max}})$ is $1$ only for responses that achieve $r_{\text{max}}$, and $0$ for all other responses, which corresponds to reward sparsity.
> - To address your concerns regarding the too-strong title and unclear theoretical backing, we have made the following two modifications:
> 	1. We changed the title of Section 3.2 from "Reward-Conditioned Policies Resolve the Limitations" to "Reward-Conditioned Policies Learn From the Full Spectrum of Response Quality". This new title reflects the property of the reward-conditioned policy, which leads to a natural fix for the limitations of direct alignment algorithms in Section 3.1.
> 	2. We also added theoretical results at the end of **Section 4 and in Appendix A**. Specifically, we showed that our method attains global convergence to the optimal policy and the suboptimality decays at the order of $N^{-1/2}$ ($N$ denotes the size of the reward-augmented preference dataset), which provides a theoretical justification for the strong empirical performance of the reward-augmented DPO introduced in this paper. Unlike prior works on goal-conditioned RL with supervised learning [4, 5], which typically establish weaker results such as local performance improvements or the optimization of a lower bound on $J(\pi)$, our analysis guarantees global convergence to the optimal policy. This distinction underscores the significance of integrating DPO-like methods with goal-conditioned approaches.

---

> > ### Author Response · Authors · 2024-11-27
> >
> > **Weakness 2: I don't understand the method. All datapoints have only a binary indicator that one response was preferred over the other. So how are you determining the value of  $r_w^i$ and $r_l^i$? Without some additional information, they would all be the same.**
> > - In our work, we mainly focus on RL from AI feedback. In RLAIF, judges, such as the reward model,  generate preference pairs along with quality scores. Our preference dataset is also defined as $\mathcal{D}_N=\{x^i, y_w^i, y_l^i, r_w^i, r_l^i\}_{i=1}^N$ (see Sec. 2).
> > - In practice, most of the preference datasets, such as UltraFeedback [1], HelpSteer2 [2], and ones in the iterative DPO [3] constructed using external reward models, also contain quality scores along with preference ranks. Therefore, studying preference alignment with judge scores is of practical importance.
> > - In Sec. 6.2 (Table 5), we also showed our method is compatible with the binarized preference dataset. Specifically, we applied a second round of DPO on the reward-augmented data relabeled with the implicit reward from the DPO model in the first round. The resulting model even outperforms models trained using judge scores from strong LLMs such as GPT-4, indicating that the DPO does not fully exploit the potential of the data and our method helps get more juice out of the binarized data.
> >
> > **Weakness 3: I think the accuracy of the reward model matters a lot, and PairRM and ArmoRM are both pretty good reward models from my understanding. It might be better to do a more stark separation where eg one flips the ArmoRM score and sees what happens. I think this would be interesting since LMs usually don't use the context in the way that we expect them to.**
> >
> > As per the reviewer's suggestion, we conducted additional ablations on the impact of reward model accuracies. Specifically, for the preference pairs in the UltraFeedback dataset, we swapped the quality scores of the chosen and rejected responses. Then DPO is performed on this dataset and our reward relabeled dataset to fine-tune the Qwen2-7B-Instruct model.  The results on the AlpacaEval 2.0 benchmark are shown in the following table:
> > |                   | LC Win Rate | Win Rate  |
> > |-------------------|-------------|-----------|
> > | Qwen2-7B-Instruct        | 20.93       | 18.22     |
> > | +DPO        | 21.46       | 19.35      |
> > | +Ours | **31.17**  | **27.58** |
> > | +DPO (swapped)       | 17.34       | 15.89  |
> > | +Ours (swapped, inference $g=5$) | 21.28  | 18.15 |
> > | +Ours (swapped, inference $g=10$) | 17.76  | 17.98 |
> >
> > It can be found from the table that performing DPO on the swapped dataset and reward relabeled swapped dataset leads to a performance decrease. Besides, conditioned on lower goal rewards, our method generates responses that are better than the higher-goal-conditioned ones. This behavior is expected since DPO now maximizes the implicit reward margin between the rejected and the chosen response which has a negative quality gap since the original preference order is relatively accurate.
> >
> > **Minor: I would not use "unlearning" as a term to describe the phenomenon since that has a very different meaning in ML literature.**
> >
> > Following the reviewer's suggestions, we have changed "unlearning" to "suppressed" to refer to the decrease in the probabilities of certain responses. We have added explanations in the revision Section 3.1.
> >
> > ---
> > We hope the reviewer could consider raising the score if we resolved the reviewer's concerns. We would be happy to have further discussions if the reviewer has any additional questions or comments.
> >
> > ---
> > [1] Cui et al. ''UltraFeedback: Boosting Language Models with Scaled AI Feedback.''\
> > [2] Wang et al. "HelpSteer2-Preference: Complementing Ratings with Preferences."\
> > [3] Xiong et al. "Iterative Preference Learning from Human Feedback: Bridging Theory and Practice for RLHF under KL-Constraint."\
> > [4] Yang et al. "Rethinking goal-conditioned supervised learning and its connection to offline rl."\
> > [5] Ghosh et al. "Learning to reach goals via iterated supervised learning."

---

> > > ### Author Response · Authors · 2024-11-28
> > >
> > > Dear Reviewer,
> > >
> > > Since the discussion period ends in **5** days, we would like to follow up to see if our rebuttal responses addressed the reviewer’s concerns. We would be more than happy to have further discussions if the reviewer has any additional questions or comments.
> > >
> > > Best,\
> > > Authors of Submission 8745

---

> > > > ### Author Response · Authors · 2024-11-29
> > > >
> > > > We thank the reviewer for the valuable insights and comments, which have significantly enhanced the quality of our manuscript. In our previous response, we clarified the tabular settings and objectives in Section 3 to address your **Weakness 1.1**.To address your **Weakness 1.2** that "Section 3.2 is titled way too strongly for a method that has no clear theoretical backing, *which is the reason for giving a 6 instead of an 8*", we revised the section titles and incorporated theoretical results to better motivate and support our method. Does our modification address your concerns? Besides, we clarified our method to resolve your **Weakness 2**, and added reward-flipping ablation studies as you suggested to address your **Weakness 3** about the impact of RM accuracy. We hope you could consider raising the score if we resolved your concerns and would like to have further discussions if you have any additional questions or comments.

---

> ### Comment · Reviewer_ofzy · 2024-11-30
>
> Thank you to the authors for their response.
>
> I would recommend modifying the writing around Table 1 to clearly indicate what the setting is and to also describe that you are not training anything but just conceptually formulating what the policy would be. Overall, the writing quality of the paper is not good, and I found a lot of things confusing (as I mentioned in my review). I recommend authors invest substantial time into the writing to clarify the settings in each section as well as what is a concrete result and what is merely speculation.
>
> I appreciate the authors' effort to run the additional ablation that I requested. I think this more clearly illustrates that the goal-conditioning is actually playing a role in the alignment procedure. I also appreciate the effort undertaken to add a new theorem and proof, both of which were interesting to read. However, I recommend the authors polish the presentation of the theorem more for the next revision.
>
> With all this in mind, I am keeping my score. I would have increased it to a 7 if the option was available, but I feel an 8 is too high for a paper that is not well-written.

---

> > ### Author Response · Authors · 2024-11-30
> >
> > We thank the reviewer for the valuable suggestions and are delighted that the reviewer is satisfied with our new ablation and theory results. In our next revision, we will modify Table 1 and its caption to clearly indicate the tabular settings considered, polish the presentation of the theoretical results, and add the above discussions to Sections 3 and 4.

---

### Official Review · Reviewer_weiM · 2024-11-09

**Soundness:** 3
**Presentation:** 2
**Contribution:** 2
**Rating:** 6
**Confidence:** 4

**Summary:**

The paper introduces a novel method to enhance the direct preference alignment of Large Language Models (LLMs) by incorporating reward-augmented data. The authors identify limitations in existing direct alignment algorithms, such as overfitting and unlearning of high-quality rejected responses, and propose a reward-conditioned policy to address these issues. The method involves a simple data relabeling technique that conditions preference pairs on quality scores, creating a reward-augmented dataset. This approach is shown to improve the performance of various models across multiple benchmarks, including instruction-following and academic benchmarks.

**Strengths:**

- The paper presents a creative solution to the problem of overfitting and unlearning in direct preference alignment by introducing reward-conditioned policies. This approach is novel and extends the capabilities of existing alignment algorithms.
- The experimental results demonstrate significant performance improvements across a range of models and benchmarks. The comprehensive ablation studies further validate the effectiveness of the proposed method.
- The proposed method has broad implications for improving the alignment of LLMs with human preferences, which is a critical area of research for developing more reliable and user-friendly AI systems.

**Weaknesses:**

- While the method is innovative, it builds heavily on existing direct preference alignment algorithms.
- The experiments are extensive but primarily focus on instruction-following and academic benchmarks. Including additional real-world applications, diverse datasets could strengthen the generalizability claims.

**Questions:**

- What are the benefits of keeping the rejected high-reward samples? Can the author provide evidence for consequential effects of the unlearning problem when LLM finetuned with the original DPO? The authors might need to start with a model that is not DPO fine-tuned and start to fine-tune using DPO and the proposed method.
- How does the proposed method compare with other recent advancements in preference alignment, such as those using model-based reinforcement learning or other reward-based approaches?
- Can the authors analyze which are rejected but high-reward samples in human feedback cases? Are they actually beneficial?

---

> ### Author Response · Authors · 2024-11-27
>
> We thank the reviewer for identifying our work's novelty, soundness, and technical contributions. The valuable comments have helped us improve our manuscript (marked **blue** in the revision). Below are our specific responses to the questions raised by the reviewer:
>
> **Weakness 1: While the method is innovative, it builds heavily on existing direct preference alignment algorithms.**
>
> We appreciate the reviewer for finding our method innovative. We would like to emphasize that the key motivation of this paper is to address the limitations of direct preference alignment algorithms, such as the indiscrimination between responses of varying quality and the difficulty of generalizing effectively to the sparse optimal responses. On the contrary, for PPO-style alignment algorithms, preference data is used to fit the reward value, which directly plays a role during the RL optimization and thus avoiding drawbacks inherent to direct alignment methods (see Section 5 for a detailed discussion).
>
> **Weakness 2: The experiments are extensive but primarily focus on instruction-following and academic benchmarks. Including additional real-world applications, diverse datasets could strengthen the generalizability claims.**
>
> The goal of RLHF and RLAIF is to enable LLMs to follow instructions aligned with human intent. Accordingly, our experiments adhere to standard setups commonly used in alignment research, focusing primarily on instruction-following benchmarks such as AlpacaEval, MT-Bench, and Arena-Hard-Auto. Additionally, we evaluate our method on a variety of academic benchmarks. These evaluations encompass diverse real-world use cases, including generating human-preferred responses and solving mathematical problems. If the reviewer has specific benchmarks for real-world applications in mind, please let us know and we would be happy to incorporate evaluations.
>
> **Question 1: What are the benefits of keeping the rejected high-reward samples? Can the author provide evidence for the consequential effects of the unlearning problem when LLM finetuned with the original DPO? The authors might need to start with a model that is not DPO fine-tuned and start to fine-tune using DPO and the proposed method.**
> We thank the reviewer for the great question. The following experiment results and discussions have been added to **Appendix B.3** in the revision.
> - Following the reviewer's suggestions, we conduct additional experiments to show the benefits of learning from the rejected high-reward samples. Specifically, using the UltraFeedback dataset, we construct two reward-augmented preference datasets by filtering out augmented data based on rejected responses with low and high reward values, respectively, which we denote as **Ours-filter-low** and **Ours-filter-high**. Compared to our method, these datasets isolate the impact of excluding low- and high-reward rejected responses as goals. The results on AlpacaEval 2.0 are shown as follows:
>
> |           | LC Win Rate | Win Rate  |
> |-----------|-------------|-----------|
> | Qwen2-7B-Instruct | 20.93        | 18.22      |
> | +DPO        | 21.46        | 19.35     |
> | +Ours        | 31.17        | **27.58**      |
> | +Ours-filter-high  | 29.36   | 27.04 |
> | +Ours-filter-low  | **31.81**   | 27.28 |
>
> It can be observed that learning from rejected high-reward samples   (**Ours** and **Ours-filter-low**) demonstrates superior performance compared to the approach that excludes these samples (**Ours-filter-high**), demonstrating the benefits of keeping the rejected high-reward samples.
> - Following the reviewer's suggestions, we conducted an additional ablation study to show the effects of the unlearning problem on the LLM that is not DPO fine-tuned. Specifically, we fine-tune Zephyr-SFT [1] with DPO and our method. We report the log probability of the high-quality rejected responses in the test set of UltraFeedback as follows:
>
> | Quality Score        | 5 | 6 | 7 | 8 |9 | 10 |
> |-------------------|-------------|-----------|-------------|-----------|-------------|-----------|
> | Zephyr-SFT  | -316        | -322     |-379|-307|-87|-41|
> | +DPO        | -436        | -466     | -478 | -398 | -266 | -75|
> | +Ours        | -321        | -314     |-383|-332|-119|-55|
> - It can be observed from the above table that the probability of high-reward responses after DPO significantly decreases compared to the SFT model and our method. Besides, the resulting model of our method also outperforms the DPO model by a considerable margin on the AlpacaEval 2.0 benchmark:
>
> |                   | LC Win Rate | Win Rate  |
> |-------------------|-------------|-----------|
> | Zephyr-SFT        | 6.21        | 3.94      |
> | +DPO        | 11.60       | 8.58      |
> | +Ours | **16.66**   | **13.37** |

---

> > ### Author Response · Authors · 2024-11-27
> >
> > **Question 2: How does the proposed method compare with other recent advancements in preference alignment, such as those using model-based reinforcement learning or other reward-based approaches?**
> > We compared with various reward-based baselines in experiments, including DPA [2], SteerLM [3], and RPO [4]. Following the reviewer's suggestions, we compared with additional reward-based approaches, including NCA-preference, NCA-reward, InfoNCA-preference, and InfoNCA-reward [5]. The results on the AlpacaEval 2.0 benchmark are shown as follows:
> >
> > |                   | LC Win Rate | Win Rate  | Avg. Len. |
> > |-------------------|-------------|-----------|-----------|
> > | Zephyr-SFT        | 6.21        | 3.94      | 893       |
> > | SteerLM    | -    | 8.21   | 1585      |
> > | DPO        | 11.60       | 8.58      | 1240      |
> > | DPA       | 11.13       | 10.58     | 1671      |
> > | NCA-preference       | 11.50       | 8.43      | 1287      |
> > | NCA-reward       | 12.87       | 9.56      | 1364      |
> > | InfoNCA-preference       | 13.68       | 11.00      | 1449      |
> > | InfoNCA-reward       | 14.83       | 11.34      | 1338      |
> > | Ours | **16.66**   | **13.37** | 1812      |
> >
> > We observe that our method outperforms these baselines by a considerable margin. The above experiment results have been added to **Figure 4 and Appendix B.3** in the revision. If the reviewer has specific model-based RL methods or other reward-based approaches in mind, please let us know and we would be happy to incorporate evaluations for them.
> >
> > **Question 3: Can the authors analyze which are rejected but high-reward samples in human feedback cases? Are they actually beneficial?**
> > - We analyze the preference dataset HelpSteer2 [6] that contains human-annotated fine-grained quality scores. In the $10162$ pairs of responses, $2778$ of them have inconsistent preference over these fine-grained quality aspects (i.e., the scores of the rejected responses are not strictly dominated by the chosen ones). This suggests that for many paired responses (~$27$ percent), human annotators do not have a clear preference for them. For example, the chosen response may be more helpful, but their correctness scores can be lower than the rejected ones. Unlearning such *similar-quality* rejected responses may hurt the model's performance.
> > - We also provide an example in the HelpSteer2 dataset. When the prompt is "write a joke", the rejected response is a joke followed by some explanations and the definition of "knock-knock" jokes, while the chosen response is another joke without explanations. Although the human annotators decided that the chosen response was more helpful, the intellectual depth score of the rejected response was higher. Our method can help learn from the entire spectrum of response quality by relabeling the data with goals such as "high helpfulness score" and "high intellectual depth score".
> >
> > ---
> > We hope the reviewer could consider raising the score if we resolved the reviewer's concerns. We would be happy to have further discussions if the reviewer has any additional questions or comments.
> >
> > ---
> > [1] https://huggingface.co/alignment-handbook/zephyr-7b-sft-full \
> > [2] Wang et al. "Arithmetic Control of LLMs for Diverse User Preferences: Directional Preference Alignment with Multi-Objective Rewards."\
> > [3] Dong et al. "SteerLM: Attribute Conditioned SFT as an (User-Steerable) Alternative to RLHF."\
> > [4] Nvidia "Nemotron-4 340B Technical Report."\
> > [5] Chen et al. ''Noise Contrastive Alignment of Language Models with Explicit Rewards.''\
> > [6] Wang et al. "HelpSteer2-Preference: Complementing Ratings with Preferences."

---

> > > ### Author Response · Authors · 2024-11-28
> > >
> > > Dear Reviewer,
> > >
> > > Since the discussion period ends in **5** days, we would like to follow up to see if our rebuttal responses addressed the reviewer’s concerns. We would be more than happy to have further discussions if the reviewer has any additional questions or comments.
> > >
> > > Best,\
> > > Authors of Submission 8745

---

> > > > ### Comment · Reviewer_weiM · 2024-11-29
> > > > **Response to authors**
> > > >
> > > > Thank you for your response, which addresses most of my concerns. I believe that this method mainly works for well SFT-ed models and high-quality DPO responses. However, the benefit of having rejected high-reward samples is still unclear. In the author's example, the request is “write a joke,” and the rejected high-reward response is the joke with an explanation. If we do not lower the probability of the rejected response enough (like using DPO), how can we ensure that the model will not respond like this in the future?
> > > >
> > > >
> > > > I appreciate the author's effort in the additional experiments; thus, I increase the score to 6.

---

> > > > > ### Author Response · Authors · 2024-11-29
> > > > >
> > > > > We thank the reviewer for the valuable comments and for raising the score! Our method is designed to augment the same dataset as DPO uses and, like DPO, does not require high-quality responses. Instead, our method is motivated to address the limitations of DPO. One limitation is that DPO "unlearns" the rejected response (i.e., decreases its probability) even if its quality is comparable to the chosen one. For instance, consider a "write a joke" prompt where the rejected response includes both the joke and an explanation (e.g., the definition of a "knock-knock" joke). While the rejected response may score lower in helpfulness, but may score higher in intellectual level. The overall scores of the responses depend on the coefficient of these two attributes. When the helpfulness attribute dominates, the model learns to avoid jokes with explanations, yet it can still generate such jokes when conditioned on maximizing intellectual-level scores. Unlike DPO, our method leverages the full spectrum of response qualities. In cases where the attributes are equally weighted, the chosen and rejected responses may have similar overall quality, indicating that useful learning signals exist in both. If the reviewer has any additional questions or comments, we are happy to have further discussions!

---

### Meta-Review · Area_Chair_rFKv · 2024-12-29

**Metareview:**

The paper introduces a novel data relabeling method for LLM preference alignment that conditions training on explicit reward scores rather than just binary preferences. This approach helps prevent "unlearning" of high-quality rejected responses and enables better generalization to optimal responses by learning patterns across the full quality spectrum. The empirical results show consistent improvements over standard DPO across multiple models and benchmarks, including both instruction-following tasks and academic evaluations.

Strengths emphasize on simplicity, as it requires only data relabeling without algorithm changes and comprehensive empirical validation across diverse models and benchmarks, and theoretical analysis providing convergence guarantees. The main weaknesses include heavy reliance on reward labels which may be noisy in practice, especially with human annotators vs GPT-4; lack of human evaluation studies to validate the real-world effectiveness; and the writing quality and clarity of explanations could be improved in several sections. Overall, there were a mix of reviews and the authors managed to address some of the concerns but given no strong champion for the work, I recommend the authors incorporate the reviewer suggestions on including human evaluation studies (QA benchmarks are very different from long-from generation). The paper may also benefit from more discussion of limitations and failure cases, and clearer positioning relative to methods using absolute preferences.

**Additional Comments On Reviewer Discussion:**

See above.

---

### Decision · Program_Chairs · 2025-01-22

Reject